# EGFR/SRC/ERK-stabilized YTHDF2 promotes cholesterol dysregulation and invasive growth of glioblastoma

Runping Fang[1,8], Xin Chen[2,8], Sicong Zhang[2,8], Hui Shi[1,8], Youqiong Ye[3], Hailing Shi[4,5,6], Zhongyu Zou [4,5], Peng Li[1], Qing Guo[2], Li Ma[2], Chuan He [4,5,6,7 ✉] & Suyun Huang[1,2 ✉]

Glioblastoma (GBM) is the most common type of adult malignant brain tumor, but its molecular mechanisms are not well understood. In addition, the knowledge of the disease-associated expression and function of YTHDF2 remains very limited. Here, we show that YTHDF2 overexpression clinically correlates with poor glioma patient prognosis. EGFR that is constitutively activated in the majority of GBM causes YTHDF2 overexpression through the EGFR/SRC/ERK pathway. EGFR/SRC/ERK signaling phosphorylates YTHDF2 serine39 and threonine381, thereby stabilizes YTHDF2 protein. YTHDF2 is required for GBM cell proliferation, invasion, and tumorigenesis. YTHDF2 facilitates m$^6$A-dependent mRNA decay of *LXRA* and *HIVEP2*, which impacts the glioma patient survival. YTHDF2 promotes tumorigenesis of GBM cells, largely through the downregulation of LXRα and HIVEP2. Furthermore, YTHDF2 inhibits LXRα-dependent cholesterol homeostasis in GBM cells. Together, our findings extend the landscape of EGFR downstream circuit, uncover the function of YTHDF2 in GBM tumorigenesis, and highlight an essential role of RNA m$^6$A methylation in cholesterol homeostasis.

[1] Department of Human and Molecular Genetics, Institute of Molecular Medicine, Massey Cancer Center, Virginia Commonwealth University, School of Medicine, Richmond, VA 23298, USA. [2] Department of Neurosurgery, The University of Texas MD Anderson Cancer Center, Houston, TX 77030, USA. [3] Department of Biochemistry and Molecular Biology, The University of Texas Health Science Center at Houston McGovern Medical School, Houston, TX 77030, USA. [4] Department of Chemistry, The University of Chicago, Chicago, IL 60637, USA. [5] Howard Hughes Medical Institute, The University of Chicago, Chicago, IL 60637, USA. [6] Institute for Biophysical Dynamics, The University of Chicago, Chicago, IL 60637, USA. [7] Department of Biochemistry and Molecular Biology, The University of Chicago, Chicago, IL 60637, USA. [8] These authors contributed equally: Runping Fang, Xin Chen, Sicong Zhang, Hui Shi. ✉email: chuanhe@uchicago.edu; suyun.huang@vcuhealth.org

Glioblastomas (GBM; World Health Organization grade IV glioma) are a devastating type of brain tumor, with a median survival of 14 months, regardless of treatment[1]. The tumor cells that invade adjacent brain tissue and contribute to the intracranial dissemination of GBM, represent a major clinical problem. The proliferation and invasion capacities of glioblastomas are a consequence of the interplay of multiple layers of altered molecular regulations. The most prevailing genetic alterations are the amplification and mutation of *EGFR* that occur in more than 50% of GBMs[1]. EGFRvIII, lacking part of the extracellular region, is a frequent activating oncogenic mutant[2]. EGFR activation by amplification, mutation, and ligand-binding in an autocrine or paracrine manner can accelerate GBM formation and is associated with poor prognosis. EGFR signaling has been reported to confer global effects on the modulation of multiple pathways in GBM. However, the effects of EGFR activation on mRNA modification are less known.

N[6]-methyladenosine (m[6]A) modification is the most prevalent internal modification existing on eukaryotic mRNA[3]. We and others have recently shown that aberrant mRNA m[6]A modifications actively assist the tumorigenicity of glioblastoma stem cells[4,5]. m[6]A-modified mRNA can be preferentially recognized by the members of the human YTH domain family, the 'reader' proteins, such as YTHDF2[6,7]. Among the mammalian YTH family proteins, YTHDF2 has been shown to destabilize m[6]A-marked transcripts through deadenylation by the CCR4–NOT complex, or through endoribonucleolytic cleavage, with the help of HRSP12-RNase P/MRP[7–9]. Depletion of Ythdf2 leads to mouse embryonic lethality due to the failure of neural stem/progenitor cell proliferation and differentiation[10,11]. Despite the important roles of YTHDF2 in brain development and function, it is unknown whether this reader is presented in GBM and affects GBM propagation.

Dysregulated cholesterol metabolism by cancer epigenetic regulators is another hallmark of GBM[12]. Cholesterol metabolism is under the transcriptional control of sterol regulatory element-binding proteins (SREBPs) and liver X receptors (LXRs). SREBPs promote cholesterol synthesis and enhance the uptake of extracellular cholesterol[13]. EGFR mutation in GBM has been shown to increase the PI3K-dependent activation of SREBP1[14]. LXRs induce cholesterol efflux by regulating the expression of apolipoprotein E (ApoE), a major apolipoprotein in the CNS that mediates the transport of cholesterol, and its transporters ABCA1 and ABCG1. LXRs also suppress cholesterol uptake by enhancing the degradation of LDLR[15,16]. Although LXR agonists have been reported to cause GBM regression[12,14], the regulatory mechanisms for LXRs expression in GBM remain unclear.

In the current study, we investigate m[6]A YTH readers in GBM, and observe a correlation between increased expression of *YTHDF2* and decreased survival of glioma patients. We identify a regulatory role of EGFR activation in YTHDF2 overexpression and elucidate the underlying mechanisms in GBM cells. We also determine the biological consequences of EGFR-mediated induction of YTHDF2 on the tumorigenicity of GBM cells. Mechanistically, we uncover that YTHDF2 accelerates m[6]A-dependent *LXRA* mRNA degradation, thereby promoting cholesterol dysregulation in GBM cells. YTHDF2 also promotes the mRNA degradation of *HIVEP2* that is critical to GBM tumorigenesis.

## Results

**YTHDF2 expression is increased in GBMs and correlates with poor prognosis.** After showing the vital role of m[6]A erasers in GBM development[5], we are interested in the roles of m[6]A readers, especially those of YTH domain family proteins, which impact diverse biological processes, in glioma tumorigenesis. We first queried The Cancer Genome Atlas (TCGA)[17,18], REMBRANDT[19], French[20], Kawaguchi[21], and Paugh[22] datasets, and found the expression of *YTHDF2*, but not *YTHDF1*, *YTHDF3*, *YTHDC1*, or *YTHDC2* correlates with poor overall survival of glioma patients in all of the datasets (Fig. 1a–c, Supplementary Fig. 1a-b). Moreover, the expression of *YTHDF2* in GBMs increased as compared with normal brains in TCGA and REMBRANDT datasets (Fig. 1d).

We then examined the protein levels of YTHDF2 in human glioma specimens. YTHDF2 shows higher protein expression in tumor relative to adjacent non-tumor tissue (Supplementary Fig. 1c). Notably, the expression of YTHDF2 protein is significantly increased in GBM and correlates with tumor grade (Fig. 1e, f). In cultured glioma tumor cell lines and normal human astrocytes, YTHDF2 protein is highly expressed in GBM cells, especially in GBM-derived stem cells (GSCs), as compared with lower-grade glioma cells (Hs683 and SW1783) and normal human astrocytes (Fig. 1g and Supplementary Fig. 1d). Moreover, YTHDF2 protein expression is reduced upon induced differentiation of GSCs (Supplementary Fig. 1e). Collectively, these results suggest that YTHDF2 might be involved in the malignancy of glioma.

**YTHDF2 regulates GBM cell proliferation, invasion, and tumorigenesis.** To examine the role of YTHDF2 in GBM tumorigenesis, we used two distinct short hairpin RNAs targeting the 3′UTR (shYTHDF2#1) or the CDS (shYTHDF2#2) of *YTHDF2* mRNA to ablate YTHDF2 expression in GSC11 and GSC7-2 cells (Supplementary Fig. 2a). Loss of YTHDF2 in the GSC cells inhibited cell proliferation, DNA replication, and invasiveness (Fig. 2a–c, Supplementary Fig. 2b, c) in vitro, whereas overexpression of YTHDF2 in LN229 cells did the opposite (Supplementary Fig. 2d-g). Moreover, the ablation of YTHDF2 in the cells reduced tumor growth of GSC11 and GSC7-2 cells (Fig. 2d, Supplementary Fig. 2h, i) as determined by in vivo intracranial tumor assay. Immunohistochemical staining of mouse brain sections revealed that YTHDF2 ablation decreased Ki-67 expression and increased cleaved Caspase-3 expression (Fig. 2e, f, Supplementary Fig. 2j). Moreover, YTHDF2-depleted tumors displayed distinct margins with significantly reduced invasive fingers of tumor (Fig. 2g). Furthermore, depletion of YTHDF2 prolonged the survival of tumor-bearing mice (Fig. 2h, Supplementary Fig. 2k). Most importantly, adding back a shYTHDF2#1-resistant YTHDF2 to the YTHDF2-depleted GSCs rescued the malignant phenotypes of GSCs (Fig. 2d–h, Supplementary Fig. 2h-k), suggesting that YTHDF2 is important for GBM tumor growth and invasion. In contrast, overexpression of YTHDF2 in LN229 cells enhanced the tumor growth and invasiveness and reduced the survival of tumor-bearing mice (Fig. 2i–k). Collectively, these results suggest that YTHDF2 regulates proliferation, invasion, and tumorigenicity of GBM cells.

**EGFR/SRC/ERK signaling sustains YTHDF2 expression.** To determine which signaling pathway drives the upregulation of YTHDF2 in GBM, the most commonly aberrantly activated pathways that include AKT, FGFR, PDGFRα/β, WNT, TGFβR1, EGFR, SRC, and ERK1/2 were tested by using inhibitors MK-2206, AZD4547, CP673451, Wnt-C59, SB-431542, gefitinib, SU6656, and SCH772984, respectively. We found the kinase inhibitors for EGFR, SRC, and ERK1/2 consistently suppressed YTHDF2 protein expression in GSC11 and GSC7-2 cells in a 24-h period (Fig. 3a, Supplementary Fig. 3a). Further analysis showed that either activation of wild-type EGFR by EGF or induced expression of EGFRvIII could promote YTHDF2 expression

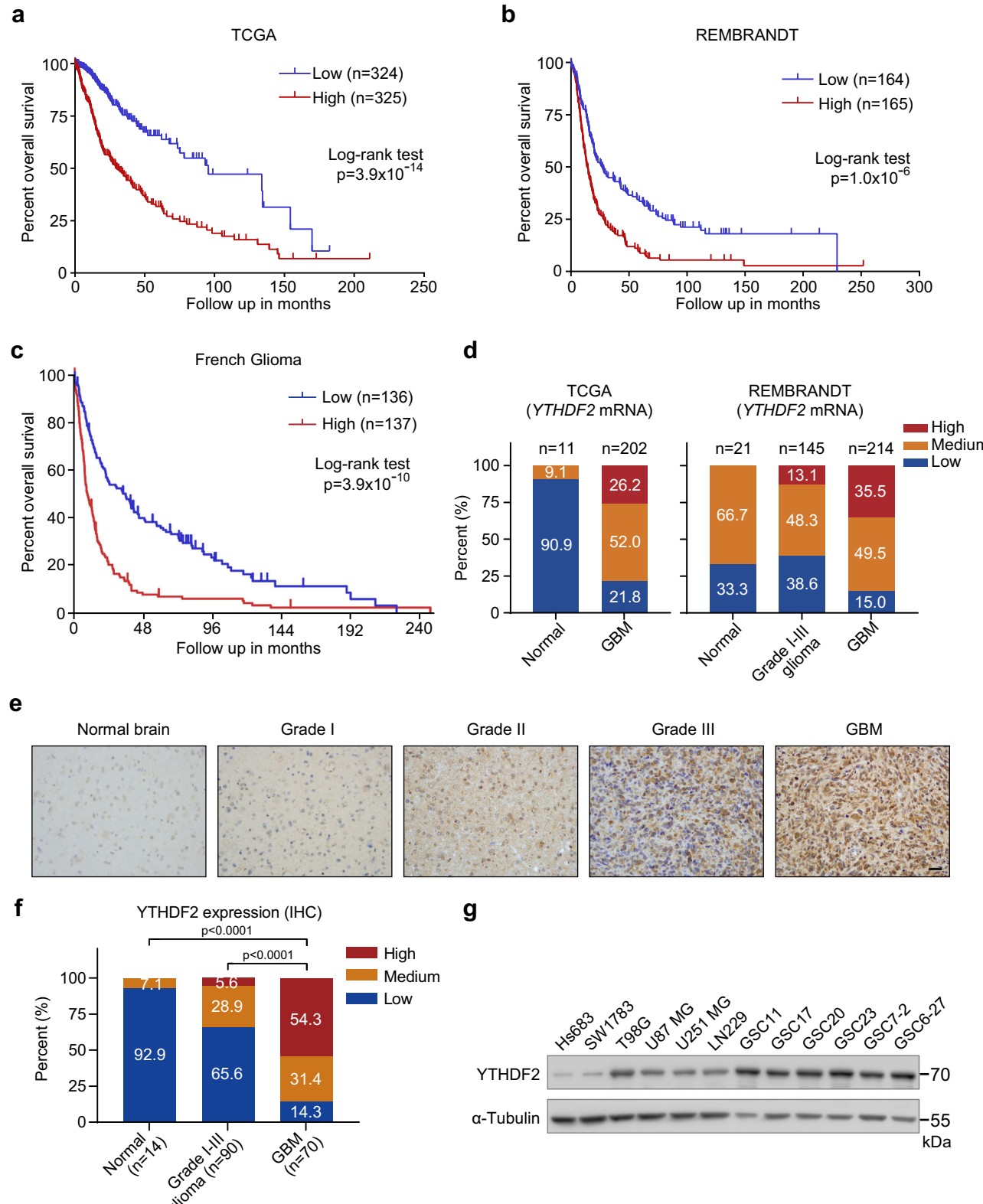

(Fig. 3b, c). Interestingly, EGFR inhibition by gefitinib or depletion by siRNA reduced the expression of YTHDF2 protein without affecting the mRNA levels (Supplementary Fig. 3b, c), suggesting a post-transcriptional regulation involved in the increase of YTHDF2. Moreover, inhibition of EGFR by gefitinib in GSC11 cells accelerated degradation of YTHDF2 protein (Fig. 3d), whereas induction of EGFRvIII expression in LN229/

EGFRvIII tet-on (doxycycline-inducible EGFRvIII) cells by doxycycline (Dox) stabilized YTHDF2 (Fig. 3e), suggesting that EGFR plays a vital role in YTHDF2 protein stabilization. Notably, hyperactivation of EGFR, which is widely found in most of GBMs including the GSCs in our current study, can activate Src[23], and ERK1/2 can be downstream of EGFR and Src[24]. Indeed, induction of EGFRvIII expression in LN229/EGFRvIII tet-on cells by

**Fig. 1 YTHDF2 is highly expressed in glioblastoma and predicts poor prognosis of glioma patients. a** Kaplan–Meier overall survival plot showing survival rates for lower-grade glioma and GBM patients having *YTHDF2* low expression (blue), and high expression (red) in the TCGA PanCancer Atlas dataset (http://www.cbioportal.org) (two-sided log-rank test). **b** Kaplan–Meier overall survival plot showing survival rates for glioma patients having *YTHDF2* low expression (blue), and high expression (red) in REMBRANDT dataset (two-sided log-rank test). **c** Kaplan–Meier overall survival plot showing survival rates for glioma patients having *YTHDF2* low expression (blue), and high expression (red) in French dataset (two-sided log-rank test). **d** Profile of *YTHDF2* mRNA expression in normal, GBM, or grade I–III glioma patients in TCGA and REMBRANDT datasets. **e** Analysis of YTHDF2 protein expression in 2 TMAs with 14 normal cores, 10 grade I astrocytomas cores, 47 grade II astrocytomas cores, 33 grade III astrocytomas cores, and 34 GBM cores; and in 36 GBM tissues, using immunohistochemical analysis. Scale bar = 100 μm. Representative image of two independent experiments. **f** Summary of YTHDF2 protein expression profile in (**e**). IHC staining score 0–1, low; 2–8, medium; 9–12, high (Kruskal–Wallis test and the post hoc Dunn's multiple comparisons test). **g** Western blotting of YTHDF2 in glioma cells (Hs683, SW1783), GBM cells (T98G, U87 MG, U251 MG, LN229), and GSCs (GSC11, GSC17, GSC20, GSC23, GSC7-2, GSC6-27). Representative blot of two independent experiments. Source data are provided as a Source data file.

doxycycline activated Src and ERK1/2 as shown by increased p-Src (Y419) and p-ERK1/2 levels (Fig. 3c). Given that YTHDF2 expression can also be regulated by Src and ERK1/2, we investigated whether EGFR-dependent YTHDF2 protein stabilization requires Src and/or ERK1/2. Treatments of the GSC11 with Src inhibitor (SU6656) or ERK1/2 inhibitor (SCH772984) inhibited the basal level and EGFR activation-induced YTHDF2 expression (Fig. 3f), indicating that EGFR's effect on YTHDF2 is mediated by Src or ERK1/2. To dissect the regulatory hierarchy (Supplementary Fig. 3d), different Src mutants that include constitutively active Src Y527F and kinase defective Src K295R were expressed in GSC11 followed by EGFR or ERK1/2 inhibition. We found that the reduction of YTHDF2 by EGFR inhibitor gefitinib in GSC11 was restored by expression of the constitutively active mutant Src Y527F but not by kinase defective K295R Src (Supplementary Fig. 3e). However, Src Y527F failed to restore the expression of YTHDF2 upon ERK1/2 inhibition, suggesting an essential role of ERK1/2 (Supplementary Fig. 3e). In agreement of this finding, EGFRvIII expression failed to stabilize YTHDF2 when ERK1/2 was inhibited by SCH772984 (Fig. 3g). Together, these results indicate that EGFR/SRC/ERK cascade is responsible for the elevated YTHDF2 protein in GBM cells.

**ERK1/2 phosphorylates YTHDF2 at serine 39 and threonine 381 to stabilize YTHDF2.** We found that endogenous YTHDF2 can associate with active ERK1/2 determined by reciprocal immunoprecipitation assays, which was further enhanced by the addition of EGF (Fig. 3h, i, Supplementary Fig. 3f). To determine whether this association also occurs in vivo, we performed a proximity ligation assay (PLA) that confirmed the interaction between YTHDF2 and ERK1/2, with an enhanced effect by EGF stimulation or EGFRvIII expression which is consistent with our in vitro observation (Fig. 3j). Next, we asked whether ERK1/2 phosphorylates YTHDF2 since the ERK1/2 kinase inhibitor SCH772984 suppressed YTHDF2 expression (Fig. 3f). As predicted by GPS 3.0 (http://gps.biocuckoo.cn/) (Supplementary Fig. 3g), several serine and threonine sites in YTHDF2 may be potentially phosphorylated by ERK1/2. Among these serine and threonine sites, S39 and T381 phosphorylations have been documented in PhosphoSitePlus® which extracts high-throughput data from published papers (https://www.phosphosite.org/homeAction.action) (Supplementary Fig. 3h). In addition, we found that S39 and T381 of YTHDF2 are conserved across different mammalian species (Supplementary Fig. 3i), suggesting that they are most likely the ERK1/2 phosphorylation sites. To examine this hypothesis, we conducted an in vitro kinase assay using recombinant ERK1 and GST purified YTHDF2. We found that wild-type YTHDF2 could be directly phosphorylated by ERK1, whereas both S39A and T381A mutants showed compromised phosphorylation (Fig. 4a). Notably, S39A/T381A double mutation completely abrogated phosphorylation by ERK1 (Fig. 4a). In line with these

observations, S39A or T381A mutant alone only slightly attenuated the association between YTHDF2 and ERK1/2, but the S39A/T381A double mutant results in a complete loss of the association (Supplementary Fig. 4a). Using the pan-phosphothreonine and -phosphoserine antibodies, we also confirmed that S39 or T381 of YTHDF2 was phosphorylated in GSC11 cells (Supplementary Fig. 4a). On the other hand, inhibition of the ERK downstream kinase MNK1 by CGP 57380 failed to affect YTHDF2 phosphorylation at these sites (Supplementary Fig. 4a), suggesting YTHDF2 as a direct target of ERK1/2.

To assess the biological effect of the YTHDF2 phosphorylation, we measured the exogenous FLAG-tagged YTHDF2. S39A/T381A mutant displayed an accelerated degradation and a shortened half-life from 13.45 to 6.71 h, compared with the wild type (Fig. 4b). Moreover, when S39A/T381A mutant was expressed to a similar amount as wild-type YTHDF2 (judged by the similar mRNA levels, Supplementary Fig. 4b), it displayed minimal effects on the promotion of invasiveness of LN229 cells (Fig. 4c), proliferation of GSC11 and GSC7-2 cells (Fig. 4d, e), and tumor growth (Fig. 4f, Supplementary Fig. 4c) compared with wild-type YTHDF2. Furthermore, we found that EGFR inhibition in GSC11 cells decreased threonine and serine phosphorylations of YTHDF2, while Dox-induced EGFRvIII expression in LN229 cells increased the phosphorylations (Supplementary Fig. 4d). Finally, a correlation between YTHDF2 protein expression and the expressions of p-EGFR (Y1173), p-Src (Y419), and p-ERK1/2 in human gliomas also suggested the importance of this EGFR/SRC/ERK/YTHDF2 cascade in tumor development (Supplementary Fig. 4e, f).

**Identification of YTHDF2 targets by high-throughput RNA-seq and RIP-seq.** It has been reported that YTHDF2 is a "reader" for m6A modification, and executes its function by destabilizing its target mRNAs in the cytoplasm or preventing demethylation of 5′UTR of stress-induced transcripts in the nucleus[7]. To investigate the downstream targets of YTHDF2, we performed RNA immunoprecipitation sequencing (RIP-seq) in GSC11 cells for YTHDF2 associated RNAs, which identified 3940 significantly enriched transcripts (Fig. 5a, and Supplementary Data 1). Then, we retrieved the m6A methylomes of GSC11 cells by MeRIP-seq from our previous study[5] and found that 3518 genes overlapped in both MeRIP-seq and RIP-seq (Supplementary Data 2), regarding as m6A-dependent targets of YTHDF2. In parallel, we performed a gene expression analysis by RNA-seq that showed 1412 differentially expressed genes in YTHDF2-knockdown GSC11 cells using two different siRNAs (Fig. 5b and Supplementary Data 3). Since YTHDF2 functions in the clearance of m6A-modified RNAs, we combined the results from RNA-seq and RIP-seq, and found 243 transcripts associated with YTHDF2 were upregulated in YTHDF2-knockdown GSC11 cells as compared with the control (Supplementary Data 4). Among them, the

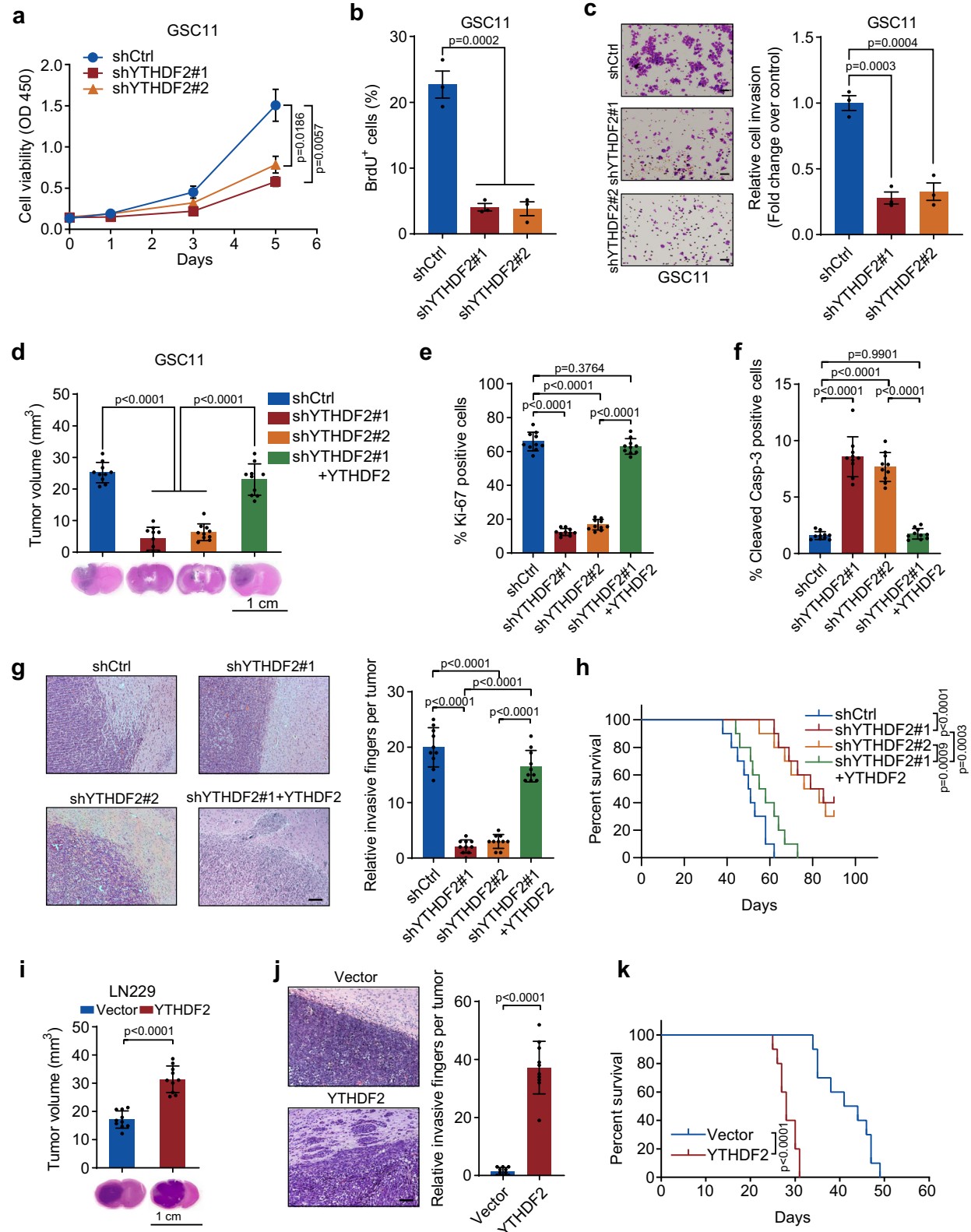

expression levels of *ATOH8, C11ORF71, CELSR3, DGCR5, HIVEP2, HSPA12A, ITPK1, KIAA1755, LXRA, PCDHA1, PRDM11, PRNP, RBSN, RNF103, RWDD2A, SIRPA, SIRT3, USP54,* and *ZBTB47* were negative correlated (spearman correlation < −0.3) with *YTHDF2* in TCGA LGG and GBM datasets (Fig. 5c, Supplementary Data 5), suggesting that these genes could be downstream effectors of YTHDF2 in glioma. Thus, we

searched the literature for the functions of these 19 genes. *LXRA* (synonym: *NR1H3*) has been reported to be a player in regulating glioblastoma cell invasion[25], and LXR agonist has been shown to promote glioblastoma cell death[12,14]. *HIVEP2* level is known to decreases in higher grade of glioma and most significantly in GBM. Transfection of HIVEP2 into glioma cells inhibited their cell growth[26]. However, the functions of the rest 17 genes in

**Fig. 2 YTHDF2 regulates GBM cell proliferation, invasion, and tumorigenesis. a** Cell viability of shCtrl and shYTHDF2 GSC11 cells was measured.
**b** Proliferation of shCtrl and shYTHDF2 GSC11 cells was assessed by 5-bromo-2′-deoxyuridine (BrdU) incorporation for 3 h. The percentage of BrdU
positive cells was quantified. **c** In vitro invasion assay for shCtrl and shYTHDF2 GSC11 cells. Representative invasion images of the cells (left panel) and
quantitation of relative cell invasion (right panel). Scale bar = 50 μm. **d** Nude mice intracranial tumor assay using shCtrl, shYTHDF2, and shYTHDF2 plus
shRNA-resistant-YTHDF2 GSC11 cells. Brain sections stained with hematoxylin and eosin (H&E) show representative tumor xenografts. Tumor volumes
were calculated. **e, f** Quantification of Ki-67 (**e**) or cleaved Caspase-3 (**f**) positive cells in brain tumor sections of shCtrl, shYTHDF2, and shYTHDF2 plus
shRNA-resistant-YTHDF2 GSC11 cells. **g** In vivo invasion assay for shCtrl, shYTHDF2, and shYTHDF2 plus shRNA-resistant-YTHDF2 GSC11 cells.
Representative H&E staining showing edges of the mice brain tumors (left panel), and the relative invasive fingers per tumor were counted (right panel).
Scale bar = 200 μm. **h** Overall survival of mice injected with the indicated GSC11 cells. **i** Nude mice intracranial tumor assay using LN229 cells with vector
only or with YTHDF2 overexpression. Brain sections stained with H&E show representative tumor xenografts. Tumor volumes were calculated. **j** In vivo
invasion assay for LN229 cells with vector only or with YTHDF2 overexpression. Representative H&E staining showing edges of the mice brain tumors (left
panel), and the relative invasive fingers per tumor were summarized (right panel). Scale bar = 200 μm. **k** Overall survival of mice injected with the
indicated LN229 cells. For **a–c**, data are mean ± S.E.M., n = 3 biologically independent experiments (one-way ANOVA Tukey's post hoc test). For **d–k**, n =
10 mice per group examined over two independent experiments. For **d–g**, data are mean ± S.D. (one-way ANOVA Tukey's post hoc test). For **h** and **k**, two-
sided log-rank test is conducted. For **i** and **j**, data are mean ± S.D. (unpaired two-sided t test). Source data are provided as a Source data file.

glioma have not been reported so far. Collectively, the above
information suggested that *LXRA* and *HIVEP2* could be direct
targets of YTHDF2 which related to GBM cell proliferation and
invasion (Fig. 5d). Indeed, the mRNAs of *LXRA* and *HIVEP2*
have several YTHDF2 binding sites (Fig. 5e, f), and the expression
of *LXRA* and *HIVEP2* negatively correlated with *YTHDF2*
expression in TCGA data (Supplementary Fig. 5a). We therefore
focused on *LXRA* and *HIVEP2* for their potential roles in med-
iating YTHDF2-induced GBM proliferation and invasion.

Next, we performed YTHDF2 cross-linking immunoprecipita-
tion (CLIP) assay to confirm the direct interactions between
*LXRA* and *HIVEP2* mRNAs and YTHDF2 protein. *LXRA* and
*HIVEP2* mRNAs could interact with YTHDF2 protein in both
GSC11 and GSC7-2 cells (Fig. 5g, Supplementary Fig. 5b),
suggesting that *LXRA* and *HIVEP2* are direct targets of YTHDF2.
Moreover, knockdown of YTHDF2 increased mRNA levels of
*LXRA* and *HIVEP2* in GSC11 and GSC7-2 cells, validating our
RNA-seq results (Fig. 5h, Supplementary Fig. 5c). In contrast,
knockdown of YTHDF2 did not affect mRNA levels of *LXRB*
(synonym: *NR1H2*), a close family member of *LXRA* and the
predominant LXR subtype in GBM[12] (Fig. 5h, Supplementary
Fig. 5c). On the other hand, overexpression of YTHDF2
decreased *LXRA* and *HIVEP2* mRNA levels in LN229 cells
without affecting *LXRB* (Fig. 5i). Importantly, in the TCGA
glioma dataset, we found that a low *LXRA* and *HIVEP2* co-
expression predicts worse prognosis of glioma patients compared
with a high co-expression (Fig. 5j). In patients with a high
*YTHDF2* expression, the *LXRA* and *HIVEP2* co-expression tends
to be low and vice versa, suggesting an indeed clinically important
regulation (Fig. 5k).

**YTHDF2 downregulates *LXRA* and *HIVEP2* through m$^6$A-
dependent mRNA decay.** We sought to investigate the
mechanism by which YTHDF2 regulates *LXRA* and *HIVEP2*
expression. The expressions of LXRα and HIVEP2 were low in
GSC11 cells, while YTHDF2 knockdown increased LXRα and
HIVEP2 protein levels without affecting LXRβ (Fig. 6a, Supple-
mentary Fig. 6a). LXRα and HIVEP2 protein levels were reduced
by induction of EGFRvIII expression in LN229/EGFRvIII tet-on
cells (Fig. 6b). Moreover, the decrease of LXRα and HIVEP2
protein levels in the EGFRvIII expressing LN229/EGFRvIII tet-on
cells was rescued by knockdown of YTHDF2 (Fig. 6b).

YTHDF2 consists of a C-terminal YTH domain, which
specifically binds to m$^6$A-containing RNA. We had mapped the
m$^6$A methylomes of GSC11 cells by MeRIP-seq in a previous
study[5]. Through analyzing the MeRIP-seq data, we found that
*LXRA* and *HIVEP2* transcripts have several significant m$^6$A peaks
(Fig. 6c, d). Moreover, knockdown of YTHDF2 enriched m$^6$A

methylated *LXRA* and *HIVEP2* mRNAs (Supplementary Fig. 6b),
which is consistent with the m$^6$A-dependent mRNA decay
function of YTHDF2[7]. According to the data of RIP-seq and
MeRIP-seq, there are peaks strongly located in 3′UTR of the
mRNA of *LXRA* and *HIVEP2*. We then constructed luciferase
reporter containing 3′UTR of *LXRA* or *HIVEP2*. We found that
Dox-induced EGFRvIII expression decreased the activity of 3′
UTR of *LXRA* and *HIVEP2*, while YTHDF2 knockdown could
reverse the effect (Supplementary Fig. 6d). Furthermore, knock-
down of YTHDF2 inhibited mRNA decay of *LXRA* and *HIVEP2*
as we measured the mRNA half-life of these genes by inhibition
of transcription with actinomycin D in GSC11 cells (Fig. 6e, f).
Thus, we determined whether YTHDF2 regulated *LXRA* and
*HIVEP2* expression through m$^6$A methylation. Given that
tryptophan residues at position 432, 486, and 491 located in
YTH domain have been shown to be critical for m$^6$A recognition
of YTHDF2[27,28], and are conserved across multiple vertebrate
species and YTH family members (Supplementary Fig. 6c), we
then generated m$^6$A-recognition defective YTHDF2 (YTHDF2
MUT) through W432A, W486A, and W491A triple mutations
(Supplementary Fig. 6c). We found that m$^6$A-recognition
defective YTHDF2 failed to enrich the mRNAs of *LXRA* and
*HIVEP2* (Fig. 6g). As a result, expression of the YTHDF2 mutant
failed to decrease the levels of mRNAs and proteins of *LXRA* and
*HIVEP2* compared with wild-type YTHDF2 (Fig. 6h, i). To
examine whether YTHDF2 regulates the expression of *LXRA* and
*HIVEP2* by mediating mRNA decay, we measured the mRNA
half-life of these genes in LN229 cells. Wild-type YTHDF2
accelerated mRNA decay of *LXRA* and *HIVEP2* indicating by the
shorter half-life as compared with the control (Fig. 6j, k), whereas
m$^6$A-recognition defective YTHDF2 has no effect on mRNA
decay of *LXRA* and *HIVEP2* (Fig. 6j, k). In addition, YTHDF2
knockdown failed to regulate the mRNA expression of *LXRA* and
*HIVEP2* in m$^6$A writer METTL14-depleted GSC11 cells (Supple-
mentary Fig. 6e). These results suggest that YTHDF2 regulates
*LXRA* and *HIVEP2* dependent on the binding of YTHDF2 to
m$^6$A-modified mRNA. Moreover, S39A/T381A mutations of
YTHDF2 compromised YTHDF2's ability to accelerate the
degradation of *LXRA* and *HIVEP2* mRNAs in YTHDF2-
depleted GSC11 cells (Supplementary Fig. 6f), implying EGFR/
SRC/ERK signaling might affect these m$^6$A-dependent mRNA
decays. Indeed, inhibition of EGFR, SRC, or ERK could increase
mRNAs of *LXRA* and *HIVEP2* (Fig. 6l). In addition, EGF
treatment in GSC11 cells or Dox-induced EGFRvIII expression in
LN229/EGFRvIII tet-on cells decreased the mRNA expression of
*LXRA* and *HIVEP2* (Fig. 6m, n), whereas knockdown of YTHDF2
reversed the effects of EGF and EGFRvIII (Fig. 6m, n).
Furthermore, in the YTHDF2-depleted GSC11 cells with EGF

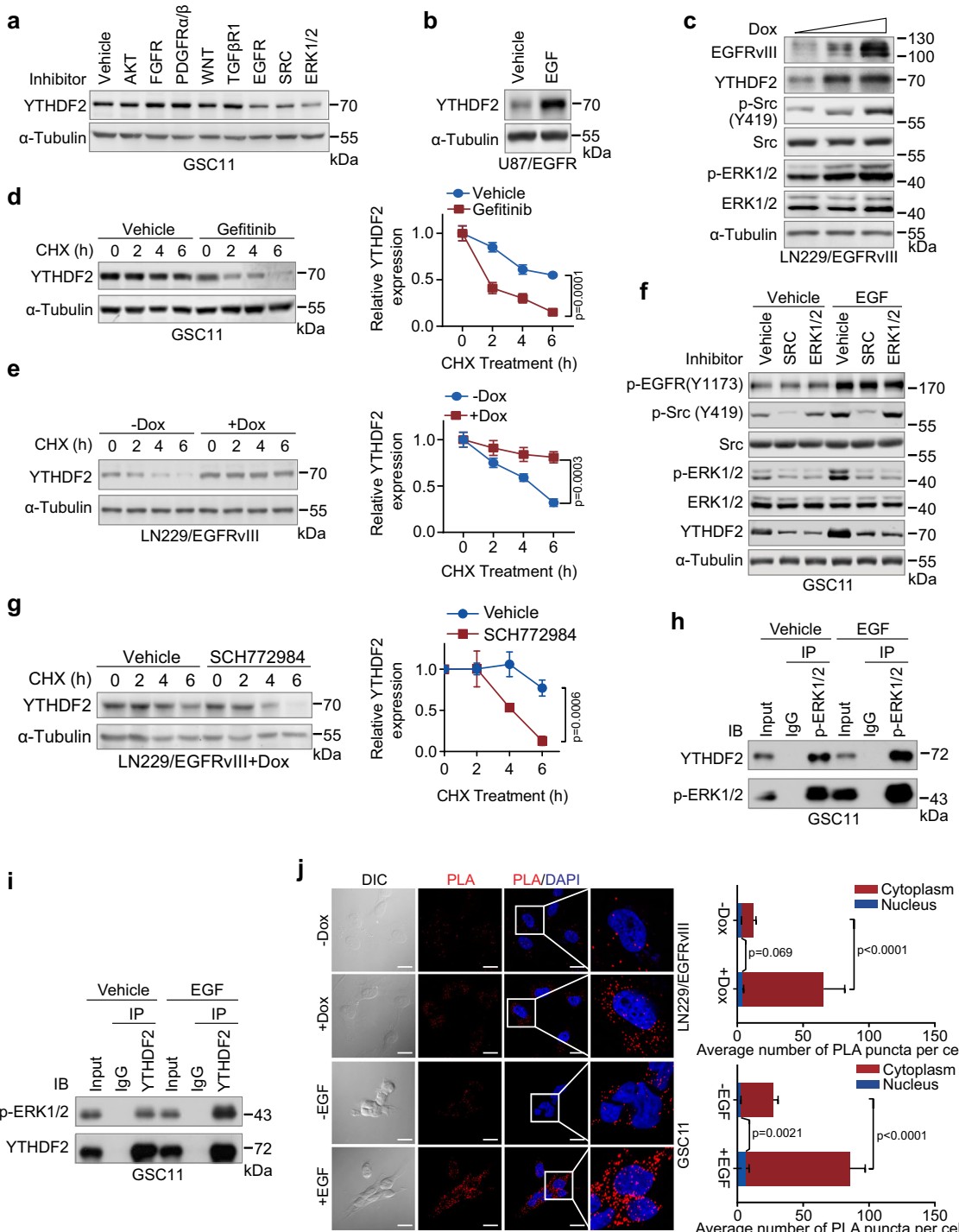

treatment, expression of wild-type YTHDF2 inhibited the mRNA expression of *LXRA* and *HIVEP2*, but m6A-recognition defective YTHDF2 did not have the effect (Supplementary Fig. 6g). Thus, the above results indicate that YTHDF2 can mediate m6A-dependent mRNA decay to restrain *LXRA* and *HIVEP2* expression under the activation of EGFR/SRC/ERK signaling in GBM cells.

**LXRα and HIVEP2 are functionally essential targets of YTHDF2 in cell proliferation, invasion, and cholesterol dys-regulation of GBM cells.** In order to determine the role of LXRα and HIVEP2 in mediating YTHDF2 biological effects on GBM proliferation and invasion, we knocked down LXRα and HIVEP2

individually or simultaneously in YTHDF2-depleted GSC11 cells. A single knockdown of LXRα or HIVEP2 partially rescued the proliferation, and the double knockdown completely restored cell viability (Fig. 7a, b, Supplementary Fig. 7a-d). In contrast, the invasion could only be rescued by knockdown of LXRα but not HIVEP2 and the double knockdown showed comparable effect as LXRα knockdown, suggesting that YTHDF2 promotes invasion through the downregulation of *LXRα* (Fig. 7c, Supplementary Fig. 7e).

LXRα regulates cholesterol homeostasis through regulating uptake and efflux of cholesterol[29]. Cancer cells depend on cholesterol for their growth but activation of LXR blocks GBM growth by reducing intracellular cholesterol[12]. Interestingly,

**Fig. 3 EGFR/SRC/ERK signaling sustains YTHDF2 expression. a** Western blotting of YTHDF2 in GSC11 cells treated with inhibitors of AKT (MK-2206), FGFR (AZD4547), PDGFRα/β (CP673451), WNT (Wnt-C59), TGFβR1 (SB-431542), EGFR (Gefitinib), SRC (SU6656), and ERK1/2 (SCH772984) for 24 h. **b** Western blotting of YTHDF2 in U87/EGFR cells cultured with or without EGF. **c** Western blotting of YTHDF2 in LN229/EGFRvIII tet-on cells treated with increasing concentration of doxycycline (Dox, 0, 1, 10 μg/mL) for 48 h. **d** Stability of YTHDF2 was measured by cycloheximide (CHX, 50 μg/mL) chase assay in GSC11 cells with vehicle or Gefitinib treatment (left panel), and analysis of YTHDF2 protein lifetime (right panel). Data are mean ± S.D. of three independent experiments (unpaired two-sided *t* test). **e** Stability of YTHDF2 was measured by CHX (50 μg/mL) chase assay in LN229/EGFRvIII tet-on cells cultured with or without Dox (10 μg/mL, left panel), and analysis of YTHDF2 protein lifetime (right panel). Data are mean ± S.D. of three independent experiments (unpaired two-sided *t* test). **f** Western blotting of YTHDF2 in GSC11 cells cultured with or without EGF and treated with inhibitor of SRC (SU6656) or ERK1/2 (SCH772984). **g** Stability of YTHDF2 was measured by CHX chase assay in LN229/EGFRvIII tet-on cells cultured with Dox plus vehicle or SCH772984 treatment (left panel), and analysis of YTHDF2 protein lifetime (right panel). Data are mean ± S.D. of three independent experiments (unpaired two-sided *t* test). **h** Co-immunoprecipitation (co-IP) of YTHDF2 with phospho-ERK1/2 (Thr202/Tyr204) in whole-cell extracts from GSC11 cells cultured with or without EGF. **i** Co-IP of phospho-ERK1/2 (Thr202/Tyr204) with YTHDF2 in whole-cell extracts from GSC11 cells cultured with or without EGF. **j** Proximity ligation assay (PLA) of phospho-ERK1/2 and YTHDF2 in LN229/EGFRvIII tet-on or GSC11 cells (left panel) and quantification of average PLA puncta per cell (right panel). Scale bar = 20 μm. Data are mean ± S.D., *n* = 6 field pictures quantified over three independent experiments (unpaired two-sided *t* test). For **a**–**c**, **f**, **h**, and **i**, representative blots of three independent experiments are shown. Source data are provided as a Source data file.

*LXRA* expression is significantly lower than *LXRB* in GBM[12]. Knockdown of YTHDF2 brought LXRα up to the level of LXRβ in GSC11 cells (Supplementary Fig. 6a). Thus, we tested whether YTHDF2 could affect cholesterol homeostasis through the increased LXRα expression. First, we measured several cholesterol metabolism-related genes regulated by LXRα. Knockdown of YTHDF2 significantly upregulated the expressions of downstream targets of LXRα, including *ABCA1*, *ABCG1*, and *APOE* (Fig. 7d, Supplementary Fig. 7f), whereas overexpression of YTHDF2 decreased the expression of these genes (Fig. 7e). Consequently, YTHDF2 depletion decreased the level of cellular cholesterol, which could be rescued by knockdown of LXRα (Fig. 7f). YTHDF2 affects cellular cholesterol in an m6A-dependent manner because overexpression of the wild-type but not m6A-recognition defective YTHDF2 increased cellular cholesterol (Fig. 7g). Moreover, cholesterol supplement could partially rescue cell viability inhibited by YTHDF2 knockdown (Supplementary Fig. 7g). LXR regulates cholesterol homeostasis by suppressing low-density lipoprotein (LDL, a carrier for cholesterol) uptake and promoting cholesterol efflux[21]. Thus, we examined LDL uptake in GSCs. YTHDF2 knockdown inhibited the uptake of LDL (Fig. 7h), and LXRα knockdown rescued LDL uptake (Fig. 7h, Supplementary Fig. 7h). Conversely, overexpression of wild-type YTHDF2, but not the m6A-recognition defective YTHDF2, increased LDL uptake (Fig. 7i, Supplementary Fig. 7i). Given that LXRα can also increase the efflux of cholesterol[22], we thus examined the role of YTHDF2 on cholesterol efflux. Knockdown of YTHDF2 increased ApoA1 dependent cholesterol efflux and knockdown of LXRα could reverse the effect (Supplementary Fig. 7j). These results suggest that YTHDF2 regulates cholesterol homeostasis through LXRα.

It has been reported that LXR regulates cell invasion through APOE[18]. To examine whether APOE plays a role in YTHDF2-mediated cell invasion, we depleted APOE gene from GSC11 YTHDF2-knockdown cells. We found that conditioned media from YTHDF2-depleted GSC11 cells reduced cell invasion, but the reduction was rescued by APOE depletion (Fig. 7j, Supplementary Fig. 7k), suggesting that YTHDF2 regulates cell invasion partially through the inhibition of APOE. Also, SSTR2, which is reported to have an anti-proliferative effect in cancer cells, is a downstream target of HIVEP2[23]. We thus examined the effect of SSTR2 knockdown in YTHDF2-mediated cell viability. We found that YTHDF2 regulated the expression of *SSTR2* in GSC11, GSC7-2, and LN229 cells (Fig. 7k, l, Supplementary Fig. 7l). Moreover, SSTR2 knockdown partially rescued the reduction of cell viability caused by YTHDF2 depletion (Supplementary Fig. 7m), suggesting that YTHDF2 regulates cell viability partially

through inhibiting SSTR2. Taken together, the above results indicate that YTHDF2 regulates GBM cell proliferation and invasion through LXRα and HIVEP2.

**Suppression of LXRα and HIVEP2 by YTHDF2 is essential for YTHDF2's function in GBM tumorigenesis.** To ascertain that suppression of LXRα and HIVEP2 by YTHDF2 is responsible for YTHDF2's function in GBM tumorigenesis, we asked whether knocking down LXRα and HIVEP2 could reverse the effects of YTHDF2 inhibition in vivo. The mice injected with GSC11- or GSC7-2-control cells all developed brain tumors resembling human GBM, whereas depletion of YTHDF2 substantially inhibited brain tumor formation (Fig. 7m). Depletion of LXRα and HIVEP2 largely abolished tumor growth inhibition by YTHDF2 knockdown (Fig. 7m) and drove the brain tumors more invasive as compared to those of YTHDF2 knockdown (Fig. 7n). Moreover, depletion of YTHDF2 prolonged the survival of tumor-bearing mice as compared with the control, whereas LXRα and HIVEP2 knockdown reversed the effect of YTHDF2 depletion on the survival of tumor-bearing mice (Fig. 7o). Furthermore, we confirmed that the control tumor xenografts have very low expression of LXRα and HIVEP2 proteins (Supplementary Fig. 7n) but depletion of YTHDF2 increased the expression of LXRα and HIVEP2 (Supplementary Fig. 7n). Taken together, these results suggest that YTHDF2 and YTHDF2-mediated regulation of LXRα and HIVEP2 play important roles in GBM tumorigenesis.

## Discussion

Ythdf2 knockout in zebrafish affects maternal-to-zygotic transition[30], and Ythdf2 mutation in mice caused female infertility[11]. Ythdf2 knockout mice also displayed early brain developmental failure due to decreased neural stem/progenitor cell proliferation and differentiation[10]. In human, YTHDF2 has been shown to affect hematopoietic stem cell amplification and leukemia survival[31,32], but our understanding of disease-associated expression and function of YTHDF2 remain limited. In this study, we identified YTHDF2 as a prognostic factor for poor glioma patient survival and found that YTHDF2 overexpression in GBM cells is required for GBM cell growth, invasion, and tumor formation in vivo. YTHDF2 knockdown inhibits GBM proliferation, invasion, and tumorigenicity largely through restraining LXRα and HIVEP2 expression which directly impacts patient survival. Furthermore, our data connected EGFR signaling, a critical driver of GBM development and therapy resistance, with the m6A-dependent mRNA clearance. EGFR-YTHDF2

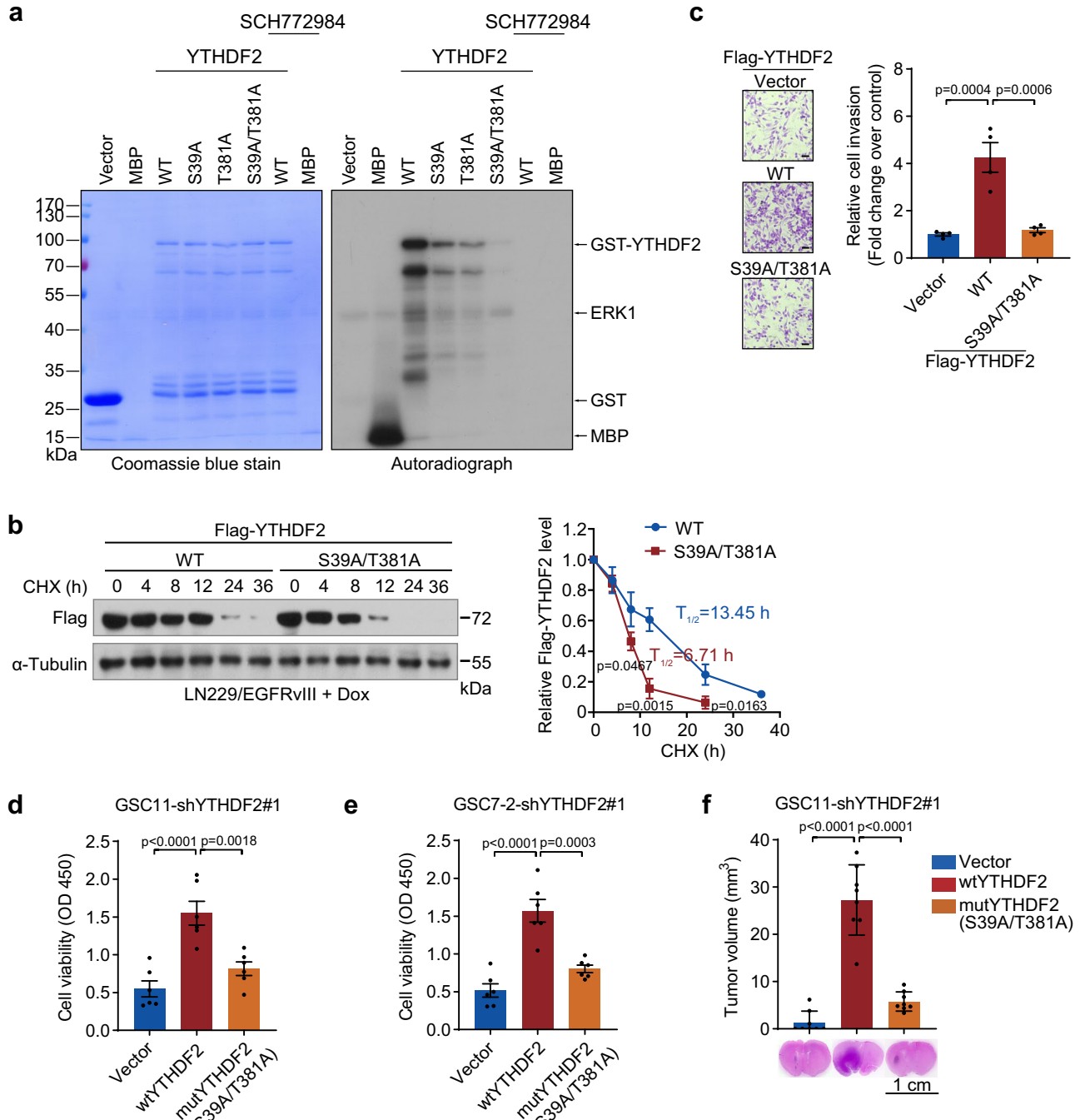

**Fig. 4 ERK1/2 phosphorylates YTHDF2 at serine 39 and threonine 381 to stabilize YTHDF2. a** In vitro kinase assay of GST-YTHDF2. Recombinant active ERK1 and wild type (WT), S39A, T381A, or S39A/T381A YTHDF2 were incubated in the presence of [γ-32P] ATP. Proteins were separated by SDS-PAGE, transferred to PVDF membrane and autoradiography performed. MBP was included as positive control and co-incubated with SCH772984 as negative control. Coomassie blue stain was shown as loading control. Representative image of three independent experiments. **b** Stability of wild type or S39A/T381A mutated Flag-YTHDF2 was measured by CHX (50 μg/mL) chase assay in LN229/EGFRvIII cells cultured with Dox (10 μg/ml) (left panel) and analysis of YTHDF2 protein lifetime (right panel). Data are mean ± S.D. of three independent experiments. (unpaired two-sided *t* test). **c** In vitro invasion assay for LN229 cells transfected with wild type or S39A/T381A mutated Flag-YTHDF2. Representative invasion images of the cells (left panel) and quantitation of relative cell invasion (right panel). Data are mean ± S.E.M., *n* = 4 biologically independent experiments (one-way ANOVA Tukey's post hoc test). Scale bar = 50 μm. **d**, **e** Cell viability of wild type or S39A/T381A mutated YTHDF2 expressing GSC11 (**e**) or GSC7-2 (**f**) cells with YTHDF2 depletion was measured by CCK-8. Data are mean ± S.E.M., *n* = 6 biologically independent experiments (one-way ANOVA Tukey's post hoc test). **f** Nude mice intracranial tumor assay using wild type or S39A/T381A mutated YTHDF2 expressing GSC11 cells with YTHDF2 depletion. Brain sections stained with H&E show representative tumor xenografts. Tumor volumes were calculated using the formula $V = ab^2/2$, where *a* and *b* are the tumor's length and width, respectively. Data are mean ± S.D., *n* = 8 mice per group examined over two independent experiments (one-way ANOVA Tukey's post hoc test). Source data are provided as a Source data file.

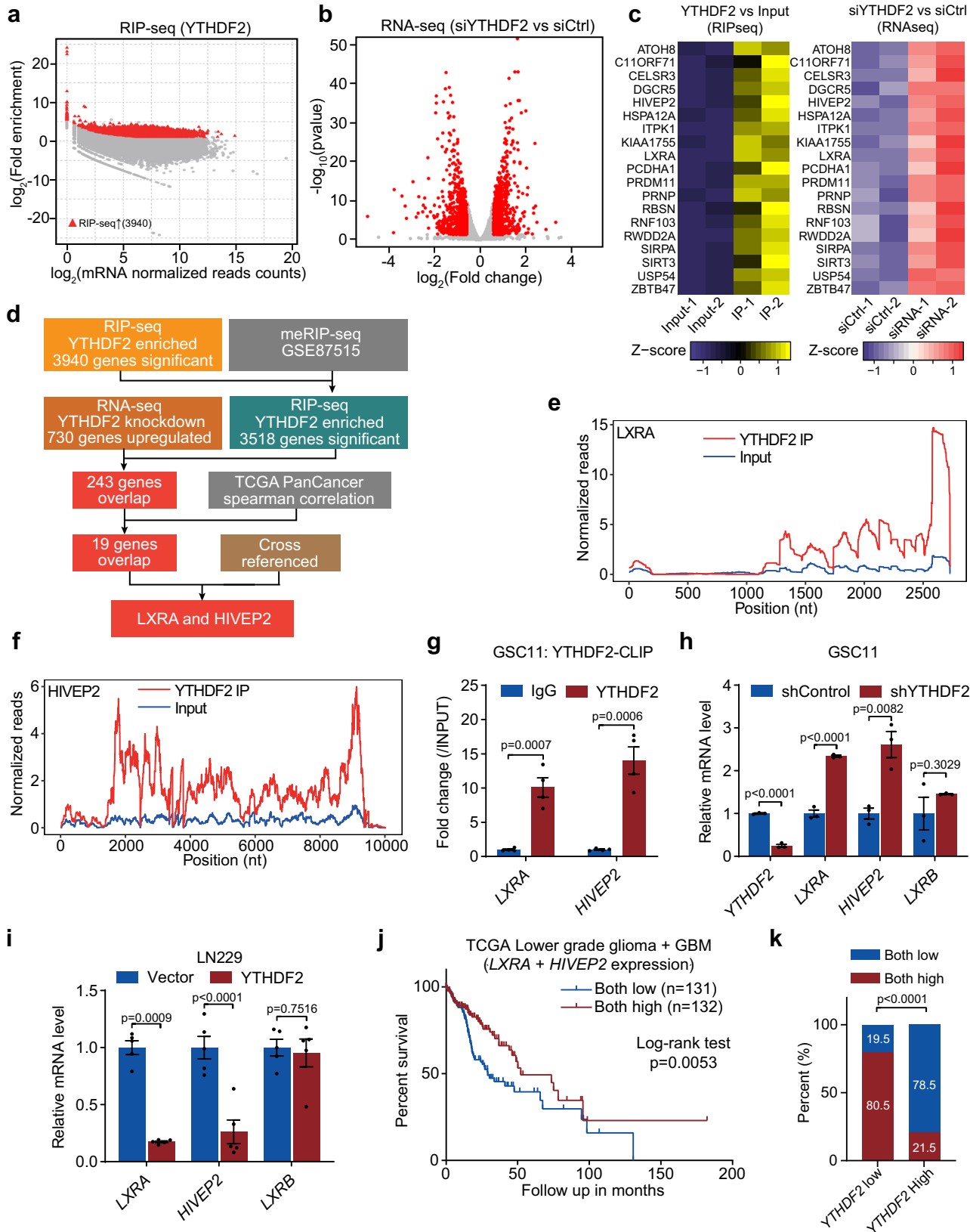

coupling represents a critical mechanism controlling the down-regulation of *LXRA* and *HIVEP2* gene expression in GBM tumorigenesis (Supplementary Fig. 8).

GBM cells are highly dependent on cholesterol for survival[29], and therefore activation of LXR subjected GBM cells to death[12,14]. *LXRA* expression is significantly lower than *LXRB* in GBM[12]. We found that YTHDF2 was highly expressed in GBM and mediated mRNA clearance of *LXRA* but did not affect the mRNA of *LXRB*. In fact, in the TCGA glioma dataset, the levels of YTHDF2 is inversely correlated with the levels of *LXRA*. Thus,

**Fig. 5 Identification of YTHDF2 targets by high-throughput RNA-seq and RIP-seq. a** Bland-Altman plot of YTHDF2 RIP-seq in GSC11 cells. For each transcript, the average signal (measured as $\log_2$ normalized reads counts) against the RIP-seq $\log_2$ fold enrichment (RIP versus INPUT) was plotted. Significantly enriched targets are highlighted in red. **b** Volcano plot of RNA-seq results in siControl and siYTHDF2 GSC11 cells. Red dots indicate fold change >1.5 and p value <0.05. **c** Heatmap of expression and enrichment of 19 overlapped genes in RNA-seq, RIP-seq, and TCGA datasets results. **d** Schematic workflow of YTHDF2 downstream targets analysis. **e, f** Plots showing YTHDF2 binding to individual mRNAs of LXRA (**e**) and HIVEP2 (**f**) genes in GSC11 cells, as measured by RIP-seq. The normalized reads distribution for input (blue) and YTHDF2 RIP along mRNAs (red). **g** CLIP-qPCR showing the association of LXRA and HIVEP2 transcripts with YTHDF2 in GSC11 cells. Data are mean ± S.E.M., n = 4 biologically independent experiments (unpaired two-sided t test). **h** Analysis of mRNA levels of YTHDF2, LXRA, HIVEP2, and LXRB in GSC11 cells stably expressing control or YTHDF2 shRNA, using qPCR. Data are mean ± S.E.M., n = 3 biologically independent experiments (unpaired two-sided t test). **i** Analysis of mRNA levels of LXRA, HIVEP2, and LXRB in LN229 cells transfected with vector or YTHDF2 expression constructs, using qPCR. Data are mean ± S.E.M., n = 5 biologically independent experiments (unpaired two-sided t test). **j** Kaplan–Meier overall survival plot showing survival rates for lower-grade glioma and GBM patients having both low LXRA expression (<median) and low HIVEP2 expression (<median) (blue), and both high LXRA expression (>median) and high HIVEP2 expression (>median) (red) in the TCGA dataset (two-sided log-rank test). **k** Percentage of low LXRA/HIVEP2 expression and high LXRA/HIVEP2 expression in YTHDF2 low expression and high expression group. Chi-squared test, $\chi^2 = 69.63$, $p < 0.0001$. Source data are provided as a Source data file.

the YTHDF2-mediated LXRA downregulation might explain the difference between expressions of LXRA and LXRB in GBM.

HIVEP2 transcriptionally regulates class I MHC, interleukin-2 receptor, somatostatin receptor II, and interferon-beta gene expressions, and regulates MYC, NF-κB, or TGF-β signaling[33]. Exogenous expression of HIVEP2 inhibited cell growth, induced differentiation, and blocked the cell cycle of glioma cells in vitro[26]. We demonstrated that YTHDF2-mediated HIVEP2 downregulation promoted GBM progression (Fig. 7). HIVEP2 is frequently downregulated in human breast cancer[34], but the mechanism is unclear. Our finding provides a possibility that YTHDF2-mediated HIVEP2 mRNA clearance accounts for inhibition of HIVEP2 expression.

Mutations of YTHDF2 might also affect the stability of YTHDF2 protein. Thus, we queried TCGA data for mutations in YTHDF2. We found that among 507 lower-grade glioma patients and 378 GBM patients, only two patients were identified by YTHDF2 mutation (one V505I mutation and one R527W mutation). These suggest that the amino acids of YTHDF2 are rarely mutated in GBM, which reinforces the importance of our findings on the regulation of YTHDF2 protein by EGFR/SRC/ERK activation.

In summary, we show that EGFR/SRC/ERK phosphorylates and stabilizes YTHDF2 protein, leading to repression of YTHDF2 target gene expression. These findings reveal the roles of EGFR signaling in tumorigenesis, as well as a mechanism responsible for YTHDF2 overexpression in GBM. We also present compelling evidence that the YTHDF2-mediated, m$^6$A-dependent mRNA clearance of LXRA and HIVEP2 is required for cholesterol dysregulation, cell proliferation, invasion, and tumorigenesis of GBM. Our data on GBM inhibition by ablation of YTHDF2 demonstrated the potential of targeting YTHDF2 for therapeutic intervention of the life-threatening brain tumor.

## Methods
**Cell lines and primary cell cultures**. Human glioma Hs683 and SW1783 cell lines and GBM T98G, U87 MG, LN229 cell line were purchased from the American Type Culture Collection (ATCC). U251 MG cell line was purchased from Sigma-Aldrich (the European Collection of Authenticated Cell Cultures). U87/EGFR cells were a gift from Dr. Frank B. Furnari. These cell lines were cultured in Dulbecco's modified Eagle's medium (DMEM) with 10% fetal bovine serum. GSC11, GSC17, GSC20, GSC23, GSC7-2, and GSC6-27 were obtained from fresh surgical specimens of human primary and recurrent GBMs under IRB protocol LAB04-0001 approved by the Institutional Review Board of the University of Texas MD Anderson Cancer Center with the written informed consents from the patients. The GSCs were cultured as tumorspheres in DMEM/F12 medium supplemented with B27 supplement (Life Technologies), bFGF and EGF (20 ng/mL each). Only early-passage GSCs were used for the study. The characteristics of the GSCs were presented previously[5]. Normal human astrocytes were purchased from LONZA (CAT#CC-2565) and cultured in AGM Astrocyte Growth Medium (CAT#CC-3186, LONZA) following the manufacturer's protocol. For Tet-inducible EGFRvIII expression, LN229/EGFRvIII cells (Tet-on) were a gift from Dr. Paul S. Mischel,

and were cultured in DMEM supplemented with 10% Tet-free FBS (Clontech). Cells were harvested and analyzed 24–48 h depending on the experimental conditions after the addition of indicated concentration of doxycycline (10 μg/mL as default concentration). For GSC differentiation, GSC11 or GSC7-2 cells were plated on tissue culture dishes in DMEM with 10% FBS for 1 week.

**Mice and animal housing**. Male and female athymic nude mice at 6–8 weeks age were used in the experiments. Mice were grouped by 5 animals in large plastic cages and were maintained under pathogen-free, ambient temperature 25 °C, humidity 48–60%, and a 12-h dark/light cycle conditions according to the NIH Guide for the Care and Use of Laboratory Animals. All mouse experiments were reviewed and approved by Institutional Animal Care and Use Committees of the University of Texas MD Anderson Cancer Center and the Virginia Commonwealth University.

**Intracranial tumor assay**. All animal studies were performed with athymic nude mice at 6–8 weeks age that were randomly allocated to each group. For the studies of investigating mice survival, mice were intracranially injected with 10,000 GSC11, 10,000 GSC7-2, or 500,000 LN229 cells. When reaching moribund condition or 90 days endpoint, mice were euthanized, and the data were analyzed by Kaplan–Meier plot. For analysis of tumor growth and invasion, mice intracranially injected with 50,000 GSC11, 50,000 GSC7-2, or 500,000 LN229 cells were euthanized 30 days after injection and brains were collected, formaldehyde-fixed and paraffin-embedded. Brain sections were processed with hematoxylin and eosin (H&E) staining to examine tumor growth and invasion. The formula of tumor volume $(V) = L \times W^2/2$, where $L$ is the length and $W$ is the width of the tumor, was used to calculate the volume of each brain tumor. Relative invasive fingers per tumor were estimated microscopically by counting protruded tumor tissue figures and disseminated areas as previously described[35].

**Analysis of RIP-sequencing data**. All samples of RNA-binding protein immunoprecipitation sequencing (RIP-seq) studies were performed by Illumina Hiseq 2500 with single-end 50-bp read length. The deep sequencing data were aligned to human reference genome version 37 (GRCh37) using hisat2[36] with default setting. After reads were mapped with hisat2, Stringtie[37] was used to calculate the read counts of each gene which represent their transcriptional expression level. Then we performed DEseq2[38] to calculate normalized reads counts in each gene and detect enriched transcripts with a double threshold on the log2-fold change (>1) and the correspondent statistical significance (P value <0.05). Analyzed data are available in Supplementary Data 1. Raw data are available under accession number GSE142828 on the NCBI GEO database.

**RNA sequencing**. Total RNA was isolated from YTHDF2 knockdown (siYTHDF2) or control (siControl) GSC11 cells using Trizol reagent (Thermo Fisher). The RNA-Seq library was generated using TruSeq Stranded mRNA Library Prep Kit (Illumina) and sequenced on a NovaSeq 6000 sequencer. Data were mapped using TopHat (TopHat v2.0.14) onto the human reference genome version 37 (GRCh37), and quantified using Cufflinks (Cufflink v2.2.1) and the GENCODE gene model. Analyzed data are available in Supplementary Data 3. Raw data are available on GEO database under the accession number GSE142828.

**mRNA stability assay**. GSCs were plated in a poly-lysine coated 6-cm dish and incubated with actinomycin D (Santa Cruz) at 5 μM for the indicated time. The first time point ($t = 0$ h) was taken as after 20 min, then 2, 4, and 6 h. Total RNA extracted from each sample was used for qRT–PCR analysis. See Supplementary Table 1 for quantitative PCR primers. According to previous studies[39], the

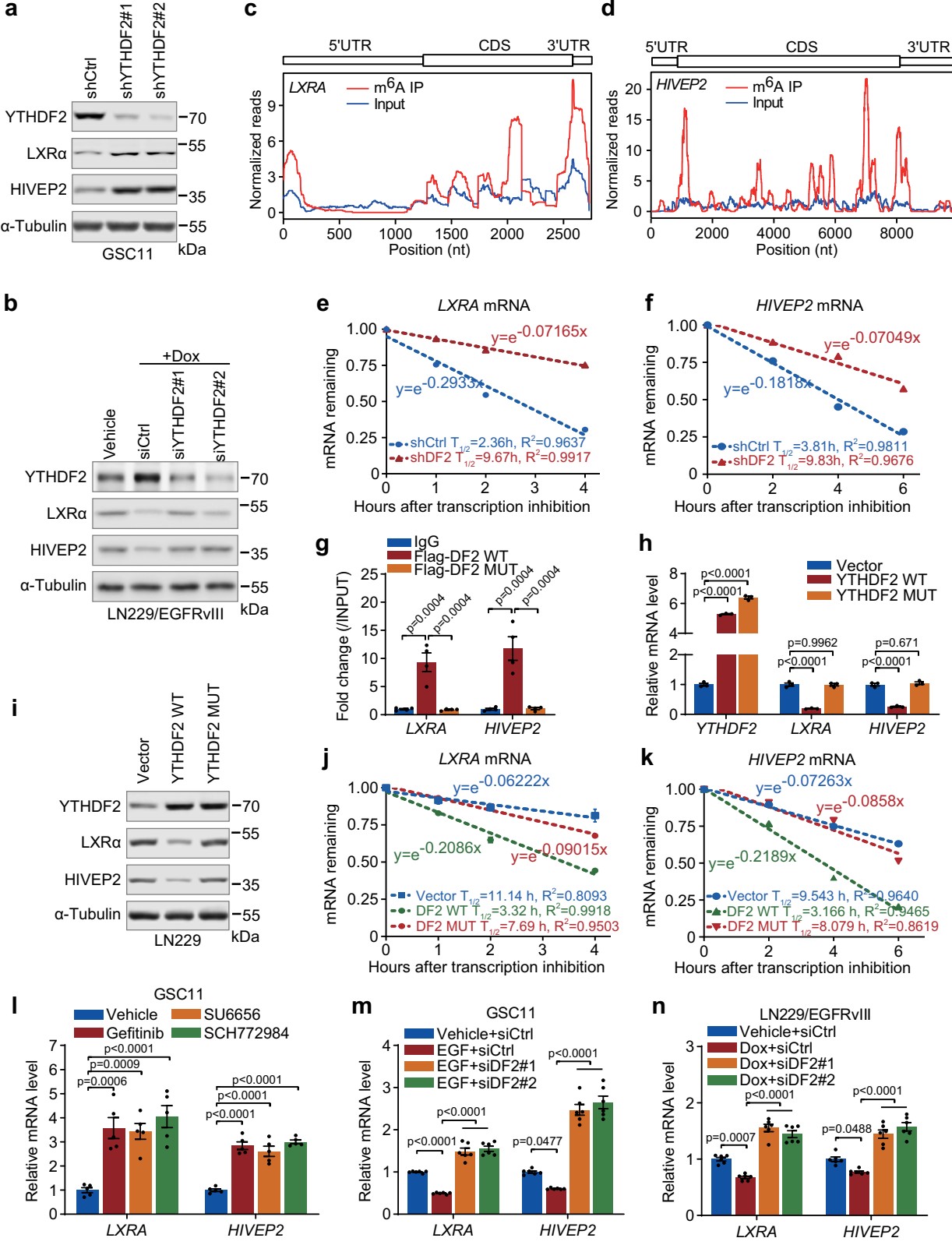

equation below was used to calculate the mRNA half-life ($t_{1/2}$):

$$t_{1/2} = \ln 2/k_{decay} \qquad (1)$$

**Plasmids and RNA interference**. YTHDF2 with N-terminal DYK tag expression pcDNA3.1 (+) plasmid was purchased from GenScript (OHu08838C). YTHDF2 W432A/W486A/W491A, S39A, T381A, and S39A/T381A mutants were generated

using QuikChange II XL Site-Directed Mutagenesis Kit (Agilent, #200522) following the manufacturer's protocols. Src Y527F and Src K295R were purchased from Addgene.

Transfections were performed using X-tremeGENE HP DNA Transfection Reagent (Roche) for plasmid and X-tremeGENE siRNA Transfection Reagent for siRNA following the manufacturer's protocols. RNAi oligonucleotides sequences are listed in Supplementary Table 2.

**Fig. 6 YTHDF2 downregulates *LXRA* and *HIVEP2* through m⁶A-dependent mRNA decay. a** Western blotting of LXRα and HIVEP2 in GSC11 cells stably transfected with control shRNA or two YTHDF2 shRNAs. **b** Western blotting of LXRα and HIVEP2 in LN229/EGFRvIII tet-on cells cultured with vehicle or Doxycycline and transfected with control siRNA or two individual YTHDF2 siRNAs. **c, d** MeRIP-seq of GSC11 cells shows m⁶A peaks at individual mRNAs of *LXRA* and *HIVEP2*. The *y*-axis shows normalized reads distribution for input (blue) and m⁶A IP (red) along the transcripts. The schematic representation of mRNA of *LXRA* or *HIVEP2* (top panel). **e, f** Lifetime of *LXRA* (**e**) and *HIVEP2* (**f**) mRNA in GSC11 cells expressing shYTHDF2 (shDF2), or shControl (shCtrl). Transcription was inhibited by actinomycin D. **g** CLIP-qPCR showing the association of *LXRA* and *HIVEP2* transcripts with wild-type (WT) or m⁶A recognition defective (MUT) Flag-YTHDF2 in LN229 cells. Data are mean ± S.E.M., *n* = 4 biologically independent experiments. **h** mRNA levels of *YTHDF2*, *LXRA*, and *HIVEP2* in LN229 cells transfected with vector, wild type (WT) or m⁶A recognition defective (MUT) YTHDF2. Data are mean ± S.E.M., *n* = 3 biologically independent experiments. **i** Western blotting of YTHDF2, LXRα, and HIVEP2 in LN229 cells transfected with vector, wild-type (WT) or recognition defective (MUT) YTHDF2. **j, k** Lifetime of *LXRA* (**j**) or *HIVEP2* (**k**) mRNA in LN229 cells expressing wild-type (WT) or m⁶A recognition defective YTHDF2 (W432A/W486A/W491A, MUT). Transcription was inhibited by actinomycin D. **l** mRNA levels of *LXRA* and *HIVEP2* in GSC11 cells treated with inhibitor of EGFR (Gefitinib), SRC (SU6656), or ERK1/2 (SCH772984). Data are mean ± S.E.M., *n* = 5 biologically independent experiments. **m** mRNA levels of *LXRA* and *HIVEP2* in GSC11 cells with EGF intervention transfected with control or two individual YTHDF2 siRNAs. Data are mean ± S.E.M., *n* = 6 biologically independent experiments. **n** mRNA levels of *LXRA* and *HIVEP2* in LN229/EGFRvIII tet-on cells cultured with or without Dox, and transfected with control or two individual YTHDF2 siRNAs. Data are mean ± S.E.M., *n* = 6 biologically independent experiments. For **a**, **b**, and **i**, representative blots of three independent experiments are shown. For **g**, **h**, and **l–n**, data were analyzed by one-way ANOVA Tukey's post hoc test. Source data are provided as a Source data file.

**In vitro kinase assay**. Recombinant GST-tagged wild type, S39A, T381A, or S39A/T381A YTHDF2 was produced in *E. coli* and purified on glutathione-Sepharose beads. Recombinant active ERK1 (CAT#E7407) and Dephosphorylated MBP (CAT#13-110) were purchased from Sigma-Aldrich. Recombinant proteins were incubated in kinase buffer (20 mM Tris–HCl, pH 7.4, 20 mM NaCl, 10 mM MgCl₂, 1 mM DTT) supplemented with 50 mM ATP (GE Healthcare) and 5 μCi [γ-³²P] ATP (PerkinElmer) for 20 min at 30 °C in the presence of 100 ng recombinant active ERK1 with or without 10 μM SCH772984. The reaction products were analyzed by SDS-PAGE electrophoresis, transferred to PVDF membrane, and autoradiographied. Finally, the PVDF membrane was stained with Coomassie Brilliant Blue R-250.

**Immunofluorescence (IF) staining**. For IF analysis of cultured cells, cells were fixed with 4% formaldehyde (Thermo Fisher) for 15 min followed by methanol permeabilization. For IF analysis of specimen, deparaffinized, rehydrated through an ethanol series followed by antigen retrieval with sodium citrate or tris-EDTA buffer according to antibody manufacturer's protocol. And then, blocked with 5% normal goat serum (Genescript) with 0.1% Triton X-100 in TBS for 60 min at room temperature. Immunostaining was performed using the appropriate primary and appropriate Alexa Fluor® 488 or Alexa Fluor® 594 secondary antibodies (Invitrogen, dilution 1:1000). Nuclei were counterstained with DAPI. Images were taken with a ZEISS Axio Scope.A1 Upright Microscope, or with a confocal imaging system (Olympus FluoView FV1000).

**Western blotting**. For evaluating protein expression in multiple cell lines, or analyzing protein expression in the cells under different conditions, protein extracts were obtained by lysing normal human astrocytes, lower-grade glioma, glioblastoma, and GSC cells, or indicated U87, LN229, or GSC cells with Laemmli sample buffer. Proteins extracted from the cells were applied equally to 10% SDS-PAGE for separation. Proteins were then transferred onto a PVDF membrane (GE), and the resulted membrane was washed and incubated with 5% Blotting-Grade Blocker (Bio-Rad) or BSA in TBST buffer for 1 h at room temperature. The membrane was washed twice with TBST and incubated with primary antibody at 4 °C overnight with gentle rotation. Then, appropriate secondary antibody conjugated with horseradish peroxidase was applied for 1 h. The chemiluminescence signals were detected with enhanced chemiluminescence (ECL) and quantified by densitometry using the ImageJ software (NIH, Bethesda, USA). At least three independent experiments have been carried out and representative results are shown.

**Antibodies**. Antibodies used in this study were the following: anti-LXRα (R&D Systems, Cat#PP-K8607-00, 1:500), anti-LXRβ (R&D Systems, Cat#PP-K8917-00, 1:500), anti-HIVEP2 (Invitrogen, Cat#PA5-100756, 1:500), anti-ERK1/2 (137F5) (Cell Signaling Technology, Cat#4695, 1:1000), anti-phospho-ERK1/2 (Thr202/Tyr204) (D13.14.4E) (Cell Signaling Technology, Cat#4370, 1:2000), anti-EGFR (D38B1) (Cell Signaling Technology, Cat#4267, 1:1000), anti-phospho-EGFR (Y1173) (53A5) (Cell Signaling Technology, Cat#4407, 1:1000), anti-Src (Cell Signaling Technology, Cat#2109, 1:1000), anti-phospho-Src (Y419) (Invitrogen, Cat#44-660G, 1:800), anti-α-Tubulin (Santa Cruz Biotechnology, Cat#sc-5286, 1:1000), anti-BrdU (Cell Signaling Technology, Cat#5292, 1:200), anti-N⁶-methyladenosine (Synaptic Systems, Cat#202003), anti-YTHDF2 (Proteintech, Cat#24744-1-AP, 1:10,000), anti-YTHDF2 antibody for RIP (Aviva Systems Biology, ARP67917_P050), anti-FLAG (Sigma-Aldrich, Cat#F1804, 1:1000), anti-SOX2 (Cell Signaling Technology, Cat#3579, 1:1000), anti-GFAP (Cell Signaling Technology, Cat#3670, 1:1000), anti-Tuji-1 (Cell Signaling Technology, Cat#4466, 1:1000),

anti-phosphoserine (Millipore, Cat#AB1603, 1:500), anti-phosphothreonine (Cell Signaling Technology, Cat#9386, 1:1000), anti-Ki-67 (Cell Signaling Technology, Cat#9664, 1:400), anti-cleaved caspase-3 (Cell Signaling Technology, Cat#9664, 1:1000), anti-rabbit IgG Alexa Fluor® 488 (Invitrogen, Cat#A-11008, 1:1000), anti-mouse IgG Alexa Fluor® 594 (Invitrogen, Cat#A-11001, 1:1000), anti-rabbit IgG HRP (Abcam, Cat#ab6721, 1:10,000), and anti-mouse IgG HRP (Santa Cruz, Cat#sc-516102, 1:5000).

**Cell viability assay**. Relative cell viability was determined by Cell Counting Kit-8 (DOJINDO). Cells were incubated 2 h after adding CCK-8 solution at 5% CO₂, 37 °C and the absorbance of each well was measured using a microplate reader (Molecular Devices) at 450 nm.

**Cell invasion assay**. Cell invasion was measured in Matrigel-coated transwell inserts containing polyethylene terephthalate filters with 8-μm pores (Millipore). Specifically, for LN229 cells, 100 μL of transiently or stably transfected 2 × 10⁵ LN229 cells in serum-free DMEM or 2 × 10⁴ GSC11 cells in growth factor-free DMEM/F12 with 0.05% bovine serum albumin were plated in the upper chamber, whereas 500 μL of DMEM (or DMEM/F12) supplemented with 10% fetal bovine serum was added to the lower well as chemoattractant. For invasion assay using conditioned culture medium, GSC11 cells were first transfected with siCtrl, siYTHDF2, siAPOE or siYTHDF2 plus siAPOE for 24 h. Then, the cells were replenished with growth factor-free DMEM/F12 medium and cultured for additional 24 h before harvesting the culture medium. In total, 2 × 10⁴ GSC11 cells were suspended with conditioned medium and plated in the upper chamber, whereas 500 μL of the conditioned medium supplemented with 10% fetal bovine serum was added to the lower well as attractant medium. The indicated cells were allowed to invade for 22 h. Non-invading cells were carefully removed with wet cotton swabs from the top of the membranes. The invading cells of the lower surface were fixed with methanol for 10 min, washed with PBS, and stained with 0.1% crystal violet for 10 min. The invading cells were either counted in at least five random fields per insert from three independent experiments, or quantified dissolved crystal violet with 33% acetic acid under 570 nm from three independent experiments. Images were taken using a Leica microscope connected to a digital camera (Leica DFC500).

**BrdU incorporation assay**. For the BrdU (5-bromo-2′-deoxyuridine) incorporation assay, cells were cultured with a BrdU-labeling reagent (Life Technologies), performed DNA hydrolysis with 1 M HCl, and stained with an anti-BrdU primary antibody (Cell Signaling) and Alexa Fluor® 594 secondary antibody according to the manufacturer's instructions.

**Proximity ligation assay**. The Proximity ligation assay was performed using Duolink™ In Situ Red Starter Kit Mouse/Rabbit (CAT#DUO92101-1KT, Sigma-Aldrich) following the manufacturer's protocol. Briefly, after permeabilization, cells were blocked against nonspecific binding with Duolink® Blocking Solution for 60 min at 37 °C. YTHDF2 (CAT#24744-1-AP, Proteintech) and phospho-ERK1/2 (CAT#5726, Cell Signaling Technology) were then incubated with cells in Duolink® Antibody Diluent overnight at 4 °C. After washed with wash buffer A, cells were incubated with Duolink® PLA Probe for 60 min at 37 °C. Cells were then incubated with the ligase and Duolink® Ligation buffer for 30 min at 37 °C. Next, cells were washed in wash buffer A for 10 min at room temperature (RT) and incubated with the polymerase and amplification buffer for 100 min at 37 °C. Finally, cells were washed in wash buffer B and 0.01× wash buffer B at RT. Then, cells were mounted with Duolink® In Situ Mounting Medium with DAPI and imaged with a confocal imaging system (Zeiss LSM700).

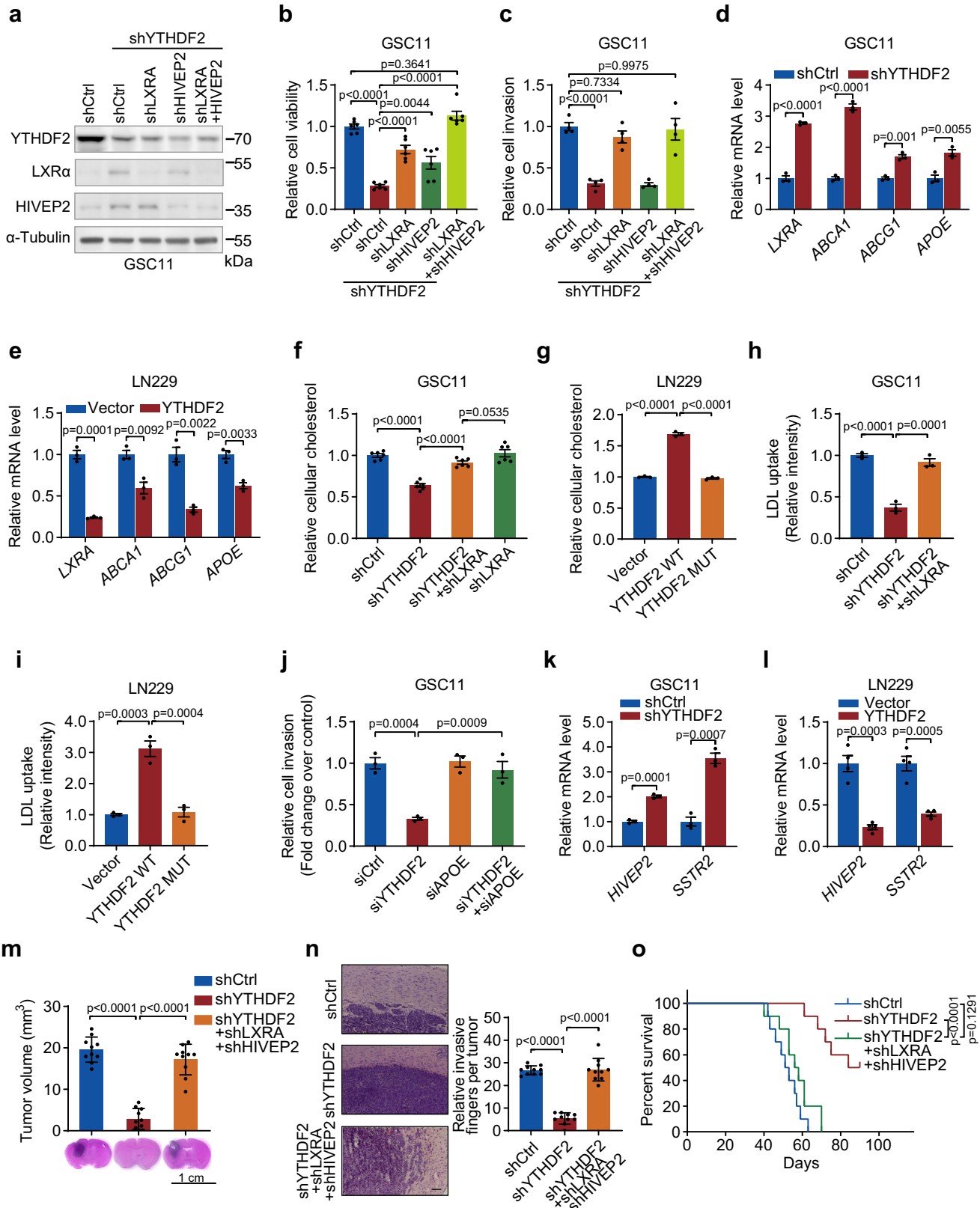

**Cellular cholesterol determination**. Cellular cholesterol levels were measured with the Amplex® Red Cholesterol Assay Kit (Invitrogen), a fluorometric technique in which cholesterol is oxidized into a ketone and hydrogen peroxide, which then reacts stoichiometrically with the Amplex Red reagent (10-acetyl-3,7-dihydrox-yphenoxazine) in the presence of horseradish peroxidase to form the fluorescent compound resorufin. To perform this assay, $5 \times 10^5$ indicated cells were plated in wells of 24-well; viable cell fractions were determined by Vi-CELL XR cell counter

(Beckman Coulter). Both viable and non-viable cells were washed in phosphate-buffered saline (PBS), resuspended at 2000 cells/µL in reaction buffer, and dispensed into wells of a 96-well tissue culture plate (Falcon/Becton Dickinson, Franklin Lakes, NJ) with 50 µL Amplex Red working solution added to each well, per the manufacturer's instructions. After incubations for 90 min at 37 °C, protected from light, fluorescence was measured on a CLARIOstar multi-mode microplate reader (BMG LABTECH) using an excitation wavelength of 530 nm

**Fig. 7 LXRα and HIVEP2 are functionally essential targets of YTHDF2 in cell proliferation, invasion, cholesterol dysregulation, and tumorigenesis of GBM cells. a** Western blotting of YTHDF2, LXRα, and HIVEP2 in GSC11 cells stably expressing shCtrl, shYTHDF2, or shYTHDF2 plus shLXRA or/and shHIVEP2. Representative blot of three independent experiments is shown. **b** Proliferation of the cells in (**a**) was measured. Data are mean ± S.E.M., $n = 6$ biologically independent experiments. **c** In vitro invasion assay for GSC11 cells expressing shCtrl, shYTHDF2, or shYTHDF2 plus shLXRA or/and shHIVEP2. Data are mean ± S.E.M., $n = 4$ wells examined over three independent experiments. **d, e** mRNA levels of *LXRA* and its downstream targets in shCtrl and shYTHDF2 GSC11 cells (**d**) or in LN229 cells expressing vector or YTHDF2 plasmid (**e**). Data are mean ± S.E.M., $n = 3$ biologically independent experiments (unpaired two-sided *t* test). **f, g** Relative cellular cholesterol of GSC11 cells stably expressing shCtrl, shYTHDF2, shLXRA, or shYTHDF2 plus shLXRA (**f**) or LN229 cells expressing vector or YTHDF2 plasmid (**g**). Data are mean ± S.E.M., $n = 6$ (**f**) or $n = 3$ (**g**) biologically independent experiments. **h, i** Quantification of LDL uptake in GSC11 cells expressing shCtrl, shYTHDF2, or shYTHDF2 plus shLXRA (**h**) or in LN229 cells expressing wild-type (WT) or m$^6$A recognition defective (MUT) YTHDF2 (**i**). Data are mean ± S.E.M., $n = 3$ biologically independent experiments. **j** In vitro invasion assay for GSC11 cells treated with conditioned culture medium from GSC11 cells transfected with siCtrl, siYTHDF2, siAPOE, or siYTHDF2 plus siAPOE. Data are mean ± S.E.M., $n = 3$ biologically independent experiments. **k, l** mRNA levels of *HIVEP2* and its downstream target *SSTR2* in shCtrl and shYTHDF2 GSC11 cells (**k**) or in LN229 cells expressing vector or YTHDF2 plasmid (**l**). Data are mean ± S.E.M., $n = 3$ (**k**) or $n = 4$ (**l**) biologically independent experiments (unpaired two-sided *t* test). **m** Nude mice intracranial tumor assay using shCtrl, shYTHDF2, and shYTHDF2 plus shLXRA and shHIVEP2 GSC11 cells. Brain sections stained with H&E show representative tumor xenografts. Tumor volumes were calculated. Data are mean ± S.D. **n** In vivo invasion assay for shCtrl, shYTHDF2, and shYTHDF2 plus shLXRA and shHIVEP2 GSC11 cells. Representative H&E staining showing edges of the mice brain tumors (left), and the relative invasive fingers per tumor were counted (right). Scale bar = 200 μm. Data are mean ± S.D. **o** Overall survival of mice injected with shCtrl, shYTHDF2, and shYTHDF2 plus shLXRA and shHIVEP2 GSC11 cells (two-sided log-rank test). For **b, c, f, g, h, i, m,** and **n**, data were analyzed by one-way ANOVA Tukey's post hoc test. For **m–o**, $n = 10$ mice per group examined over two independent experiments. Source data are provided as a Source data file.

---

and an emission wavelength of 590 nm. A cholesterol standard curve was determined for each plate using a cholesterol standard diluted at various concentrations in reaction buffer.

**Lentiviral transduction for stable cell lines.** Lentiviral vectors expressing YTHDF2 and W432A/W486A/W491A mutants were PCR amplified and cloned into pLVX-puro vector (Clontech). Lentiviral vectors expressing non-targeting pLKO.1 control shRNA (SCH002), and shRNA constructs targeting *YTHDF2* (NM_016258) shRNA#1 (TRCN0000254410) and shRNA#2 (TRCN0000265510), or targeting *LXRA* (NM_005693) shRNA#1 (TRCN0000022238) and shRNA#2 (TRCN0000438551), or targeting *HIVEP2* (NM_006734) shRNA#1 (TRCN0000236545) and shRNA#2 (TRCN0000022097) were obtained from Sigma-Aldrich. shRNA for *LXRA* and *HIVEP2* were generated according to the pLKO.1 protocol from Addgene. The lentiviral vectors were co-transfected with packaging vectors psPAX2 and pMD2.G (Addgene) into 293FT cells for lentivirus production. To establish stable cell lines, GSC cells or LN229 cells were transduced by the above lentiviruses with polybrene (8 μg/mL, Sigma-Aldrich). After 72 h of transduction, GSC cells were selected with 2 μg/mL puromycin, 5 μg/mL blasticidin, or 400 μg/mL geneticin (G418), and LN229 cells were selected with 0.5 μg/mL puromycin for 1 week and expanded as polyclonal populations. For the YTHDF2 rescue experiment, shRNA targeting 3′UTR of *YTHDF2* (shYTHDF2#1) was used for knockdown.

**LDL uptake analysis.** To analyze LDL uptake in the indicated LN229 or GSC11 cells, 2 mL of cell suspension containing $6 \times 10^4$ cells was plated in 6-well plate with cover slide. For GSC11 cells, the cells were attached to 0.01% poly-L-lysine coated cover slide. After attachment, cells were incubated with 5 μg/mL DiI conjugated LDL (DiI-LDL, Invitrogen) for 4 h at 37 °C in DMEM/F12 (GSC11) or DMEM (LN229) culture media containing 1% lipoprotein deficient serum (LPDS, Sigma-Aldrich). The incubation was stopped by washing with PBS, and the cells were fixed in 4% PFA for 15 min at room temperature. After fixation, the cells were mounted by anti-fade mountant with DAPI purchased from Life Technologies. Representative images were taken using Olympus FV1000 confocal microscope. To quantify the LDL uptake, the ImageJ software was used to analyze the average fluorescence intensity of 50 cells from each group.

**YTHDF2 RNA immunoprecipitation.** RNA immunoprecipitation was performed with the Magna RIP™ RNA-Binding Protein Immunoprecipitation Kit (Millipore) according to the manufacturer's instructions. Briefly, whole-cell lysates were prepared by freeze-thaw, cleared by centrifuging, and incubated with YTHDF2 antibody (Aviva System Biology). Bead-captured complexes were then washed extensively and eluted twice by synthetic peptide (Aviva System Biology). Then, eluates were combined for RNA extraction by Trizol. For RIP-seq, rRNA-depletion and DNase I treatment were performed on both Input and RIP samples. The relative interaction between protein and RNA was determined by qPCR and normalized to input.

**Inhibitors and ligands treatment.** For inhibitors treatment, indicated cells were treated with inhibitor of AKT (0.5 μM MK-2206), FGFR (1 μM AZD4547), PDGFRα/β (0.1 μM CP673451), WNT (10 μM Wnt-C59), TGFβR1 (10 μM SB-431542), EGFR (5 μM Gefitinib), SRC (10 μM SU6656), and ERK1/2 (10 μM SCH772984) for 24 h. For EGF treatment, U87/EGFR cells were serum-starved for

24 h and then treated with 100 ng/mL EGF treatment. After 16 h, the cells were collected and subjected to western blotting.

**Cycloheximide (CHX) chase analysis.** For endogenous YTHDF2 cycloheximide chase analysis, GSC11 cells with vehicle or 5 μM Gefitinib treatment, or LN229/EGFRvIII cultured with or without Dox, were incubated with 100 μM of cycloheximide for various times. For Flag-tagged YTHDF2 cycloheximide chase analysis, after transfection with Flag-tagged wild-type (WT)-YTHDF2 or S39A/T381A YTHDF2 for 16 h, LN229/EGFRvIII cells cultured with doxycycline, were incubated with 100 μM of cycloheximide for various times. Cell lysates were prepared and total proteins were quantified by the DC™ (detergent compatible) protein assay (Bio-Rad), and subjected to western blots. Cell lysis and Western blotting. Quantification was achieved by the ImageJ software (NIH, Bethesda, USA). Band total gray signal was quantitated and normalized to α-tubulin. The final turnover rate at each time point is the percentage of YTHDF2/α-tubulin at $t = 0$ of each experimental group.

**Co-immunoprecipitation (co-IP).** Cells were treated with EGF (100 ng/mL) or vehicle for 10 min. Then cells were lysed in co-IP buffer (10 mM HEPES pH 8.0, 100 mM NaCl, 0.1 mM EDTA, 20% glycerol, 0.2% NP-40, Halt™ Phosphatase Inhibitor Cocktail, and Halt Protease Inhibitor Cocktail (Thermo Fisher)). Lysates were centrifuged and cleared by incubation with 25 μl of Protein G agarose (Millipore) for 1.5 h at 4 °C. The pre-cleared supernatant was subjected to IP using the indicated primary antibodies at 4 °C overnight. Then, the protein complexes were collected by incubation with 30 μl of Protein G agarose (Millipore) for 2 h at 4 °C. The collected protein complexes were washed 6× with co-IP buffer and analyzed by western blotting.

**Cholesterol efflux assay.** For cholesterol efflux analysis, $2 \times 10^6$ indicated GSC11 cells were plated into 6-well and labeled with [$^3$H] cholesterol (1.0 μCi/mL) (PerkinElmer, NET139250UC) in the presence of acyl-CoA:cholesterol O-acyltransferase inhibitor (2 μg/mL) (Sigma-Aldrich, Sandoz 58-035) for 48 h (37 °C, 5% CO$_2$). After 18 h equilibration incubation, one set of wells were used to determine background efflux (efflux to serum-free media with NO acceptor). The remaining cells were washed with PBS and incubated in serum-free media in the absence or presence of ApoA-I (30 μg/mL) (Meridian Life Science, A95120H) for 4 h. Media and cells were both collected for radioactivity determination by a Beckman LS6500 scintillation counter. The rate of cholesterol efflux is determined by the following formula:

$$\%\text{Cholesterol efflux} = (\text{media counts} \times \text{dilution factor})$$
$$/([\text{media counts} \times \text{dilution factor}] + [\text{cell counts} \times \text{dilution factor}]) - \%\text{Blank efflux}$$
$$(2)$$

**MeRIP-qPCR.** GSC11 cells stably expressing YTHDF2 or control shRNA were used for total RNA extraction. Intact poly-A RNA was purified from total RNA, and subjected to denaturation at 70 °C for 10 min. Then, the denatured RNA was incubated with m$^6$A antibody in RIP buffer at 4 °C for 2 h with gentle rotation. The RNA and antibody complexes were captured by Dynabeads Protein G (Invitrogen) at 4 °C for 2 h with rotation. To elute m$^6$A containing RNA, beads were incubated twice with 6.7 mM N$^6$-methyladenosine 5′-monophosphate sodium salt in RIP buffer for 1 h each at 4 °C. The eluted m$^6$A containing RNAs were precipitated with 0.1 volumes of 3 M sodium acetate and 20 μg glycogen in 2.5 volumes of ethanol at

−80 °C overnight. Precipitated RNA was reverse-transcribed and quantified by qPCR for m⁶A enrichment.

**Immunohistochemical (IHC) staining**. Tissue microarray (BS17017b and GL808) was purchased from US Biomax (Derwood, MD, USA). Anonymous archived human glioblastoma tissue slides were obtained from The University of Texas MD Anderson Cancer Center under a protocol approved by the institutional review board with informed consent. For immunohistochemical (IHC) staining, GBM xenografts, human surgical specimens tissue slides, or TMAs were deparaffinized, rehydrated through an ethanol series followed by antigen retrieval with sodium citrate or tris-EDTA buffer according to antibody manufacturer's instruction. Sections were blocked with 5% normal goat serum (Genescript) with 3% BSA in TBS for 60 min at room temperature and were incubated with 3% $H_2O_2$ in PBS for 15 min at room temperature to block endogenous peroxidase and then incubated with appropriate primary antibodies at 4 °C overnight. IHC staining was performed with horseradish peroxidase (HRP) conjugates using DAB detection. The immunoreactive score (IRS) was calculated as, for percentage of positive cells (A): 0 = no positive cells; 1 = <10% of positive cells; 2 = 10–50% positive cells; 3 = 51–80% positive cells; 4 = >80% positive cells. For intensity of staining (B): 0 = no color reaction; 1 = mild reaction; 2 = moderate reaction; 3 = intense reaction.

$$\text{The immunoreactive score(IRS)} = A \times B \quad (3)$$

Score 0–3 = low (1st quartile), 4–8 = medium (2nd and 3rd quartiles), 9–12 = high (4th quartile).

**Analysis of YTHDF2 mRNA expression in TCGA and REMBRANDT datasets**. YTHDF2 mRNA expression data were retrieved from cBioPortal (www.cbioportal.org) and Betastasis (www.betastasis.com). YTHDF2 mRNA expressions were stratified by quartiles of all expressions: 1st quartile = low expression, 2nd and 3rd quartiles = medium expression, and 4th quartile = high expression.

**RNA isolation and quantitative real-time PCR**. For RNA isolation, TRIzol reagent (Life Technologies) was added to the indicated GSC or LN229 cells, and total RNA was extracted from the cells. mRNA was purified from the total RNA by poly-A selection with GenElute™ mRNA Miniprep Kit (Sigma). cDNAs were synthesized from the purified mRNA by using iScript™ Reverse Transcription Supermix (Bio-Rad). For quantitative real-time PCR, the PCR reactions were set with PowerUp™ SYBR® Green Master Mix (Life Technologies) on a 7500 Fast Real-time PCR System (Applied Biosystems) following the manufacturer's instruction. Primers used in this study were summarized in Supplementary Table 1.

**Luciferase reporter assay**. The 3′UTR of LXRA (ENST00000467728.5, https://uswest.ensembl.org/Homo_sapiens/Transcript/Summary?g=ENSG00000025434;r=11:47257979-47268845;t=ENST00000467728) or HIVEP2 (ENST00000012134.7, https://uswest.ensembl.org/Homo_sapiens/Transcript/Summary?g=ENSG00000010818;r=6:142751469-142945031;t=ENST00000012134) was amplified by PCR from the cDNA of GSC11 cells using specific primers and was inserted into the NheI and XhoI sites in the downstream of the luciferase gene in the pmirGLO vector. The primers used are listed in Supplementary Table 1.

For luciferase reporter assay, LN229/EGFRvIII tet-on cells ($2 \times 10^4$ cells per well in 24-well plates) were transfected with control or YTHDF2 siRNAs, and with pmirGLO luciferase reporter plasmids containing 3′UTR of LXRA or HIVEP2 using X-tremeGENE HP DNA Transfection Reagent (Roche). Luciferase activities were measured with the Dual-Luciferase Reporter Assay System (Promega) 24 h after transfection. Data were first normalized firefly luciferase activity to Renilla luciferase activity for each set of triplicate samples, and then compared to vector-only group. All experiments were performed at least three times independently.

**CLIP-qPCR**. Cross-linking and RNA immunoprecipitation (CLIP) RNA immunoprecipitation was performed as follows, following ultraviolet cross-linking, GSC11 or LN229 cells were harvested and nuclear extracts were isolated and sonicated. One microgram of YTHDF2 (Proteintech) antibody or normal rabbit IgG (Millipore) was conjugated to Protein A/G Magnetic Beads (Thermo Fisher) by incubation at 4 °C for 4 h, followed by 3× wash and incubation with pre-cleared nuclear extraction in RIP buffer (150 mM KCl, 25 mM Tris pH 7.4, 5 mM EDTA, 0.5 mM DTT, 0.5% IGEPAL® CA-630, 1× protease inhibitor) at 4 °C overnight. After washing with RIP buffer for three times, beads were resuspended in 80 μL PBS, followed by DNA digestion at 37 °C for 15 min and incubation with 50 μg of Proteinase K (Sigma-Aldrich) at 37 °C for 15 min. Input and co-immunoprecipitated RNAs were recovered by TRIzol (Invitrogen) extraction and used for qPCR analyses using primers listed in Supplementary Table 1.

**Statistical analysis**. Data are presented as the mean ± standard error of the means (SEM), or standard deviations (SD). GraphPad Prism 8.3 was used for statistical analysis, two-sided student's t test and Mann–Whitney test were used for unpaired data. Kaplan–Meier survival data were analyzed using two-sided log-rank test. The Spearman correlation test was used to assess the relationships of gene expression.

One-way ANOVA with Tukey's post hoc or Dunnett's post hoc analysis was used for multiple comparisons. p values <0.05 were considered significant.

**Reporting summary**. Further information on research design is available in the Nature Research Reporting Summary linked to this article.

## Data availability

The RNA sequencing and RIP-sequencing data are deposited at the Gene Expression Omnibus (GEO) repository under the accession number GSE142828. The Cancer Genome Atlas (TCGA) PanCancer Atlas dataset referenced during the study are available in a public repository from the cBioPortal website (https://www.cbioportal.org/), REMBRANDT dataset is available in G-DOC website (https://gdoc.georgetown.edu/gdoc), French, Kawaguchi, and Paugh datasets are available in R2: Genomics Analysis and Visualization Platform (https://hgserver1.amc.nl/cgi-bin/r2/main.cgi). All the other data supporting the findings of this study are available from the corresponding author upon reasonable request. Source data are provided with this paper.

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

## Acknowledgements

We thank The Genomic and RNA Profiling Core and The Biostatistics and Informatics Shared Resource in Baylor College of Medicine for RNA-seq and helping analyze the RNA-seq data. We thank Dr. Frank B. Furnari for U87/EGFR cells and Dr. Paul S. Mischel for tet-regulatable LN229/EGFRvIII cells. This work was supported in part by US NIH grants R01CA182684 and P30 CA016059, and by Paul M. Corman MD Chair in Cancer Research endowment fund.

## Author contributions

R.F., X.C., S.Z., and Hu.S. conceived the project and designed the research. R.F., X.C., S.Z., Hu.S., Q.G., L.M., P.L., Y.Y., Ha.S., and Z.Z. performed the research, analyzed, and interpreted data. R.F., S.Z., X.C., C.H., and S.H. wrote the manuscript. C.H. and S.H. conceived and designed the project, interpreted data, wrote the manuscript, and provided study supervision.

## Competing interests

The authors declare no competing interests.
