## [Peer Review File · Nature Communications]

Reviewers' comments:

Reviewer #1 (Remarks to the Author); expert on GBM mouse models and cholesterol metabolism:

The manuscript entitled "EGFR/SRC/ERK-stabilized YTHDF2 promotes cholesterol dysregulation and invasive growth of glioblastoma" investigates the role of the YTHDF2 protein in the context of glioblastoma, the most common primary brain tumor in adults. YTHDF2 detects N6-methyladenosine (m6A) mRNA modifications and facilitates mRNA degradation by endoribonucleolytic digestion. Based on TCGA data, high levels of YTHDF2 are correlated with a poor overall survival. The authors nicely link EGFR/SRC/ERK signaling to show that YTHDF2 is phosphorylated (Threonine 381 and Serine 39) and stabilized. YTHDF2 controls the proliferation and migration of human GBM cells. Mechanistically, YTHDF2 drives the reduction of LXRA and HIVEP2 mRNA, which in turn facilitates cholesterol metabolism in glioblastoma cells, promoting their growth and migration. Overall, this is a very compelling manuscript that has a lot of exciting data. The model systems are well balanced and appropriate. The techniques used in the manuscript are cutting-edge.

I have a couple of suggestions to improve the paper.

1. Figure 1G: Could the authors include the normal astrocytes in their analysis. In addition, would YTHDF2 levels fall when the CSC are differentiated.?

2. In order to prove the role of cholesterol in the context of YTHDF2 silencing it would be tempting to test whether or not exogenous cholesterol could rescue some of the biological effects elicited by YTHDF2 silencing, e.g. proliferation and migration.

Reviewer #2 (Remarks to the Author); expert on RNA biology and methylation:

The manuscript by Fang et al. describes an oncogenic signaling cascade linking the EGFR/SRC/ERK pathway to YTHDF2 protein stabilisation and YTHDF2-dependent regulation of mRNA decay in glioblastoma.

YTHDF2 is an m6A reader and the authors focused their study on YTHDF2 since it is specifically increased in GBMs (among 4 tested m6A readers) and it is associated with poor prognosis. Compelling evidences (siRNA, rescue, overexpression experiments in different cell lines) demonstrated that YTHDF2 regulates invasion and tumorigenesis.

Co-immunoprecipitation experiments showed that ERK interacts with YTHDF2 and that phosphorylation of YTHDF2 on two residues (S39 and T381) stabilizes YTHDF2.

YTHDF2 mRNA targets were identified by high-throughput RNA-seq (following YTHDF2 KD) and RIP-seq experiments. This led the authors to focus on two candidate mRNAs (LXRA and HIVEP2) that were found in both sets of high-throughput data. YTHDF2 binding to these 2 mRNAs, as well as YTHDF2-dependent regulation of the stability of these 2 mRNAs, were validated. An m6A-recognition defective YTHDF2 mutant was generated and failed to recognize the 2 candidate mRNAs.

In addition, regulation of invasion by YTHDF2 was clearly shown to be dependent on the regulated expression of LXRA and HIVEP2 and to their effects on cholesterol homeostasis.

Overall, this manuscript contains plenty of novel interesting data. The conclusions are well supported by the experiments.

I have a few main concerns:

-The conclusion that ERK directly phosphorylates YTHDF2 is not supported by the classical set of experiments (e.g. kinase assays with recombinant proteins...). Even if ERK is shown to interact with YTHDF2, is it possible that downstream kinases (e.g. MNK) phosphorylate YTHDF2?

-Fig 3g: the claimed increased ERK-YTHDF2 interaction upon EGF stimulation is not very obvious. Can it be confirmed with alternative methods (PLA,...)?

-What is the consequence of YTHDF2 phosphorylation on its subcellular localization? Also, can the authors use PLA approaches to determine the subcellular localization of the ERK-YTHDF2 interaction?

-Can the authors map the m6A position(s) on the LXRA and HIVEP2 mRNAs?

-Luciferase reporter constructs containing LXRA or HIVEP2 UTRs would reinforce the conclusions.

-Are the LXRA transcription targets found to be regulated in the RNAseq data (following YTHDF2 KD)?

-Can the authors be more precise on the EGFR mutational status of the GBM cell lines used in this study? Is there a correlation between EGFR mutation/amplification and YTHDF2 phosphorylation? Do EGFRi-resistant GBM cell lines exist? It would be interesting to examine the YTHDF2 phosphorylation status in such cells.

Reviewer #3 (Remarks to the Author); expert on EGFR signaling:

In this manuscript the authors describe their observations suggesting that the m6A modification reader protein YTHDF2 is engaged by EGFR activation to promote glioblastoma (GBM) cell growth properties. They outline evidence that YTHDF2 promotes GBM cell survival (their observations that YTHDF2 promotes proliferation, invasiveness and tumor growth properties are likely secondary to this survival effect), that an EGFR to src to Erk signaling cascade stabilizes YTHDF2 protein, that YTHDF2 destabilizes transcripts for LXRA and HIVEP2, and that suppression of these transcripts restores viability to cells depleted of YTHDF2. Although the conclusions of the study as stated in the abstract come across as diffuse and somewhat confusing, the strength of the study is that the authors appear to have uncovered a novel pathway downstream of activated EGFR that may contribute to the malignant properties of GBM. The major weaknesses of the manuscript are that it is poorly written and presented, conclusions are often overstated relative to the presented data, and the major conclusions are lost in a backdrop of wide-ranging observations.

Specific concerns are largely related to scientific rigor:

1. If, as fig 3 suggests, EGFRvIII regulates YTHDF2 by promoting protein stability but not transcript abundance, why do YTHDF2 transcript levels correlate with patient prognosis?
2. Phosphorylation of YTHDF2 downstream of EGFR, src and Erk is not demonstrated; the authors merely demonstrate that two putative phosphorylation sites are necessary for YTHDF2 association with Erk.
3. For the in vivo study of fig 2, Ki67 staining should be carried out on tumors to support differences observed in cell proliferation in vitro. Similarly, staining should be carried out for markers of cell death (Caspase3 or TUNEL). Staining for YTHDF2 could also be carried out to show loss of expression in tissue, as well as restored expression in the rescue cohort.
4. Details should be provided concerning in vitro invasion assays.
5. Figure panels 2a, 2e, and 2f are more appropriate for supplemental materials.
6. The impact of the S39A/T381A mutant in cell and tumor assays beyond in vitro invasiveness should be shown.
7. In fig 5 multiple distinct siRNAs should be employed in all studies for scientific rigor.
8. In fig 5 additional experiments should be carried out to demonstrate that the YTHDF2 m6A recognition mutant can rescue the EGF- or EGFRvIII-induced decrease in LXRA and HIVEP2 mRNA expression.
9. Do the YTHDF2 phosphorylation site mutants affect LXRA and HIVEP2 transcript stability?
10. In fig 6a, 6b, and 6c the authors should show the effect of the single knockdown alone, compared to the triple knockdown, as well as different permutations of double knockdowns to unravel individual contributions.
11. Again in fig 6, scientific rigor dictates the use of multiple shRNA constructs for each target.
12. For the cholesterol experiments of fig 6, the effect on cellular cholesterol with knockdown of

LXRA alone should be shown as a control.

13. Figure 6k and l suggest that SSTR2 as a downstream target of HIVEP2 has antiproliferative effects that YTHDF2 is able to regulate expression of SSTR2. The authors conclude that YTHDF2 regulates cell proliferation in this context, however, cell proliferation is never shown. These experiments need to be included.

Reviewer #4 (Remarks to the Author); expert on m6A RNA and RIP-seq:

In this manuscript, Fang et. al. report an oncogenic role of m6A reader YTHDF2 in GBM for the first time. Together with their previous study on the function of m6A demethylase ALKBH5 in GBM (Zhang et. al., Cancer cell, 2017.), it reveals the aberrant of m6A modification and its regulatory machinery plays an important role in the tumorigenesis and development of GBM. The authors also showed that the EGFR/SRC/ERK signaling phosphorylates YTHDF2 and thus promotes YTHDF2 stability, which represents a novel mechanism regulating expression of YTHDF2 at the post-translational level. Overall, this is an interesting and timely study, and the paper is well written. Nevertheless, while the authors have provided extensive data to support the mechanism and function of YTHDF2 in GBM, there are some additional experiments and analyses that would further greatly strengthen the paper.

1. The authors used S39A and T381A mutated YTHDF2 to demonstrate these two sites are important for ERK binding and YTHDF2 stability. However, the direct evidence of phosphorylation of S39 and T381 in GBM are missing. Besides, in order to further support the conclusion that ERK-mediated phosphorylation of YTHDF2 stabilizes YTHDF2, the authors could examine whether ERK inhibitor shorten YTHDF2 protein half-life in GBM cells.

2. The authors identified YTHDF2 targets by integrative analysis of RIP-seq and RNA-seq. Although 3,940 transcripts were associated with YTHDF2, only 19 genes among them were upregulated by YTHDF2 knockdown (KD). I think the number is relatively low as Wang et. al. reported that YTHDF2 had a significant and global impact on target RNA stability across the transcriptome (Wang et. al. Nature, 2014.). This is largely due to the small number (in total only 111) of differentially expressed transcripts identified by their RNA-seq data. It is better to redo RNA-seq with a higher reader coverage, and to ensure the KD efficiency of siRNAs (or it might be better to use shRNAs instead). Or, at least the authors can try combine RIP-seq and meRIP-seq first to identify the transcripts associated with YTHDF2 and also containing m6A sites as m6A-dependent target genes. Then analyze the expression changes of m6A-dependent target genes in global or in individual genes upon YTHDF2 KD.

3. Besides m6A-recognition defective YTHDF2, the authors could also knockdown m6A methyltransferase METTL3 and/or METTL14 in GBM cells to demonstrate m6A-dependent regulation of LXRA and HIVEP2 by YTHDF2.

4. In addition to the activation by EGFR/SRC/ERK, mutations in YTHDF2 amino acid sequence might also affect the stability of YTHDF2. For example, the mutation of lysine might prevent ubiquitination and degradation of YTHDF2 protein. Please check the mutation data in GBM and discuss.

POINT-BY-POINT RESPONSES

We thank the reviewers for thorough evaluation of our original manuscript. The reviewers' comments and suggestions are highly constructive and genuinely helpful. We have carefully considered all of the critiques and then performed the experiments suggested by the reviewers. After addressing all the points raised by the reviewers and revising the manuscript accordingly, including presentation of necessary details, further clarifications and new results, we feel that the revised manuscript represents a significant and strong advance over previous work.

We have incorporated the responses to reviewers into the revised manuscript and updated the figures in the revised manuscript to reflect changes we have made in response to the comments. The point-by-point responses are provided herein.

Responses to Reviewer #1

The manuscript entitled “EGFR/SRC/ERK-stabilized YTHDF2 promotes cholesterol dysregulation and invasive growth of glioblastoma” investigates the role of the YTHDF2 protein in the context of glioblastoma, the most common primary brain tumor in adults. YTHDF2 detects N6-methyladenosine (m6A) mRNA modifications and facilitates mRNA degradation by endoribonucleolytic digestion. Based on TCGA data, high levels of YTHDF2 are correlated with a poor overall survival. The authors nicely link EGFR/SRC/ERK signaling to show that YTHDF2 is phosphorylated (Threonine 381 and Serine 39) and stabilized. YTHDF2 controls the proliferation and migration of human GBM cells. Mechanistically, YTHDF2 drives the reduction of LXRA and HIVEP2 mRNA, which in turn facilitates cholesterol metabolism in glioblastoma cells, promoting their growth and migration. Overall, this is a very compelling manuscript that has a lot of exciting data. The model systems are well balanced and appropriate. The techniques used in the manuscript are cutting-edge.

Response: We thank the reviewer for acknowledging the importance and strength of our work.

I have a couple of suggestions to improve the paper.

1. Figure 1G: Could the authors include the normal astrocytes in their analysis. In addition, would YTHDF2 levels fall when the CSC are differentiated?

Response: We thank the reviewer for the suggestions. We thus analyzed YTHDF2 expressions in normal astrocytes and differentiated CSC. The normal human astrocyte cells were purchased from LONZA (CAT#CC-2565). The result from the new experiment showed that, the expression of YTHDF2 in normal human astrocyte was very low as compared to LN229 GBM cells and GSC cells (Supplementary Fig. 1d). Also, differentiated GSC11 and GSC7-2 cells have decreased YTHDF2 expression as compared to the cells in stem status (Supplementary Fig. 1e).

2. In order to prove the role of cholesterol in the context of YTHDF2 silencing it would be tempting to test whether or not exogenous cholesterol could rescue some of the biological effects elicited by YTHDF2 silencing, e.g. proliferation and migration.

Response: We thank the reviewer for the suggestions. We thus supplied cholesterol to the YTHDF2-depletion GSC11 cells by using the cholesterol-M β CD complex with the related dose tested in GBM cells as described in a previous publication (Villa et al., 2016, Cancer Cell 30, 683–693). We found that exogenous cholesterol partially rescued the inhibitory effect of YTHDF2-depletion on the proliferation of GSC11 cells *in vitro*. These results have been updated in Supplementary Fig. 7g.

Responses to Reviewer #2

Reviewer #2 (Remarks to the Author); expert on RNA biology and methylation:

The manuscript by Fang et al. describes an oncogenic signaling cascade linking the EGFR/SRC/ERK pathway to YTHDF2 protein stabilisation and YTHDF2-dependent regulation of mRNA decay in glioblastoma.

YTHDF2 is an m6A reader and the authors focused their study on YTHDF2 since it is specifically increased in GBMs (among 4 tested m6A readers) and it is associated with poor prognosis. Compelling evidences (siRNA, rescue, overexpression experiments in different cell lines) demonstrated that YTHDF2 regulates invasion and tumorigenesis.

Co-immunoprecipitation experiments showed that ERK interacts with YTHDF2 and that phosphorylation of YTHDF2 on two residues (S39 and T381) stabilizes YTHDF2.

YTHDF2 mRNA targets were identified by high-throughput RNA-seq (following YTHDF2 KD) and RIP-seq experiments. This led the authors to focus on two candidate mRNAs (LXRA and HIVEP2) that were found in both sets of high-throughput data. YTHDF2 binding to these 2 mRNAs, as well as YTHDF2-dependent regulation of the stability of these 2 mRNAs, were validated. An m6A-recognition defective YTHDF2 mutant was generated and failed to recognize the 2 candidate mRNAs.

In addition, regulation of invasion by YTHDF2 was clearly shown to be dependent on the regulated expression of LXRA and HIVEP2 and to their effects on cholesterol homeostasis.

Overall, this manuscript contains plenty of novel interesting data. The conclusions are well supported by the experiments.

Response: We thank the reviewer for the positive comments.

I have a few main concerns:

-The conclusion that ERK directly phosphorylates YTHDF2 is not supported by the classical set of experiments (e.g. kinase assays with recombinant proteins...). Even if ERK is shown to interact with YTHDF2, is it possible that downstream kinases (e.g. MNK) phosphorylate YTHDF2?

Response: We thank the reviewers for these suggestions. We have now conducted *in vitro* kinase assay using recombinant active-ERK1 and GST purified YTHDF2. We

found that purified wild type YTHDF2 can be phosphorylated by active-ERK1 *in vitro*, whereas both S39A and T381A mutants show reduced phosphorylation by active-ERK1 as compared to wild type YTHDF2 (Fig. 4a). The S39A/T381A double mutant completely abrogate YTHDF2 phosphorylation by active-ERK1 (Fig. 4a), suggesting that these are the two primary phosphorylation sites. On the other hand, MNK1 inhibition by CGP 57380 failed to suppress YTHDF2 serine or threonine phosphorylation (Supplementary Fig. 4a).

-Fig 3g: the claimed increased ERK-YTHDF2 interaction upon EGF stimulation is not very obvious. Can it be confirmed with alternative methods (PLA,..)?

Response: Following the suggestion, we used PLA to detect the interaction between ERK and YTHDF2, and found that EGF stimulation in GSC11 cells or EGFRvIII inducible expression in LN229/EGFRvIII cells could obviously increase the PLA signals, suggesting that EGF stimulation and EGFRvIII inducible expression enhanced the interaction between YTHDF2 and ERK. We have updated these results in Fig. 3j.

-What is the consequence of YTHDF2 phosphorylation on its subcellular localization? Also, can the authors use PLA approaches to determine the subcellular localization of the ERK-YTHDF2 interaction?

Response: Following the suggestions, we have used PLA approaches to determine the subcellular localization of the ERK-YTHDF2 interaction. We found that EGF stimulation or EGFRvIII expression could increase the interaction between YTHDF2 and ERK, and the interaction mostly located in the cytoplasm (Fig. 3j). Although the ERK-YTHDF2 interaction in the nucleus did increase a little bit by EGF stimulation or EGFRvIII expression, the proportion of the interaction in the nucleus vs cytoplasm was similar in control and in EGF stimulation or EGFRvIII expression groups (Fig. 3j). These results suggest that YTHDF2 phosphorylation did not specifically affect its subcellular localization.

-Can the authors map the m⁶A position(s) on the LXRA and HIVEP2 mRNAs?

Response: Yes, we have mapped the m⁶A peaks on LXRA and HIVEP2 mRNAs in Fig. 6c and d.

-Luciferase reporter constructs containing LXRA or HIVEP2 UTRs would reinforce the conclusions.

Response: We thank the reviewer for the suggestion. We constructed luciferase reporters containing the 3'UTR of LXRA or HIVEP2, because RIP-seq and meRIP-seq data showed there are YTHDF2 and m⁶A peaks strongly located in the 3'UTR of these mRNAs (Fig. 5e, f and Fig. 6c, d). We then examined the activity of these 3'UTR luciferase reporters in tet regulated LN229/EGFRvIII cells. We found that doxycycline could reduce the luciferase activities of 3'UTR constructs of LXRA and HIVEP2, while YTHDF2 knockdown reversed the effect (Supplementary Fig. 6d).

-Are the LXRA transcription targets found to be regulated in the RNAseq data (following YTHDF2 KD)?

Response: Yes, we found that LXRA transcription targets ABCA1 was upregulated by 1.50 folds, and APOE was upregulated by 1.65 folds in the RNAseq data following YTHDF2 KD (Supplementary table 3).

-Can the authors be more precise on the EGFR mutational status of the GBM cell lines used in this study? Is there a correlation between EGFR mutation/amplification and YTHDF2 phosphorylation? Do EGFRi-resistant GBM cell lines exist? It would be interesting to examine the YTHDF2 phosphorylation status in such cells.

Response: The EGFR status of the GBM cell lines used in this study is as the following. EGFR in T98G, U87 MG, LN229, U251 MG, GSC7-2, GSC6-27, GSC20 cells are wildtype, GSC23 cells contain EGFRvIII, and GSC11 and GSC17 cells have EGFR amplification.

To investigate if there is a correlation between EGFR mutation/amplification and YTHDF2 phosphorylation, we have used GSC11 cells which have EGFR amplification as well as LN229/EGFRvIII cells. We found that EGFR inhibition in GSC11 cells decreased threonine and serine phosphorylation of YTHDF2, while Dox-induced EGFRvIII expression in LN229 cells increased the phosphorylation (Supplementary Fig. 4d). These results suggest that the phosphorylation of YTHDF2 was correlated with EGFR amplification/mutation.

We have diligently searched the literature for EGFRi-resistant GBM cell lines. However, we could not find an ideal cell line which is suitable for the purpose.

Responses to Reviewer #3

In this manuscript the authors describe their observations suggesting that the m6A modification reader protein YTHDF2 is engaged by EGFR activation to promote glioblastoma (GBM) cell growth properties. They outline evidence that YTHDF2 promotes GBM cell survival (their observations that YTHDF2 promotes proliferation, invasiveness and tumor growth properties are likely secondary to this survival effect), that an EGFR to src to Erk signaling cascade stabilizes YTHDF2 protein, that YTHDF2 destabilizes transcripts for LXRA and HIVEP2, and that suppression of these transcripts restores viability to cells depleted of YTHDF2. Although the conclusions of the study as stated in the abstract come across as diffuse and somewhat confusing, the strength of the study is that the authors appear to have uncovered a novel pathway downstream of activated EGFR that may contribute to the malignant properties of GBM. The major weaknesses of the manuscript are that it is poorly written and presented, conclusions are often overstated relative to the presented data, and the major conclusions are lost in a backdrop of wide-ranging observations.

Response: We thank the reviewer for acknowledging the strength of our work. Also, we have taken the reviewer's concern on the writing, presentation, and conclusion of our manuscript very seriously. We then have carefully re-written many parts of the

manuscript to make sure the presentation and conclusions are precisely stated. Furthermore, we have performed the experiments suggested by the reviewer. The detailed results have been presented in various new/revised figures. We sincerely hope that we have adequately and satisfactorily addressed the concerns of the reviewer.

Specific concerns are largely related to scientific rigor:

1. If, as fig 3 suggests, EGFRvIII regulates YTHDF2 by promoting protein stability but not transcript abundance, why do YTHDF2 transcript levels correlate with patient prognosis?

Response: We first explored several large GBM datasets and found a correlation between YTHDF2 mRNA expression and patient prognosis, which indicates a potential biological role of YTHDF2 in GBM development. We confirmed the protein expression of YTHDF2 is also correlated with patient prognosis. However, when we compared the YTHDF2 mRNA expression with the YTHDF2 protein expression in GBM patients, we found that the positive rate of high expression of YTHDF2 protein in GBM patients was higher than that of YTHDF2 mRNA (Fig. 1d-f), suggesting that the expression YTHDF2 could be also regulated at post-transcriptional level in GBM. Thus, we went further to investigate the regulation of YTHDF2 expression by several major pathways in GBM. Unexpectedly, we found a protein level regulation by EGFR signaling. Although it is unclear how YTHDF2 transcript levels correlate with patient prognosis, we think our finding is highly significant since mRNA and protein expressions usually do not correlate well in tumors, and it is more important to focus on the protein expression.

2. Phosphorylation of YTHDF2 downstream of EGFR, src and Erk is not demonstrated; the authors merely demonstrate that two putative phosphorylation sites are necessary for YTHDF2 association with Erk.

Response: We concur with the reviewer's points. Thus, to demonstrate that phosphorylation of YTHDF2 is downstream of EGFR and Erk, we performed *in vitro* kinase experiments and *in vivo* experiments using pan-phosphothreonine or -phosphoserine antibody for detecting the phosphorylated YTHDF2.

In the *in vitro* kinase assay with recombinant active-ERK1, YTHDF2 wild-type and phospho-mutants, we found that ERK1 phosphorylates YTHDF2-WT strongly, and S39A- and T381A-YTHDF2 to a lesser extent, but not S39A/T381A double mutant YTHDF2 (Fig. 4a). These results suggest that YTHDF2 can be phosphorylated by ERK directly and the S39 and T381 sites are the primary ERK phosphorylation sites.

Next, in the *in vivo* experiment using pan-phosphothreonine and -phosphoserine antibodies, we detected threonine and serine phosphorylation of YTHDF2 in GSC11 cells (Supplementary Fig. 4d). Moreover, the threonine and serine phosphorylation of YTHDF2 was suppressed by an EGFR inhibitor in the cells (Supplementary Fig. 4d). Also, these phosphorylations of YTHDF2 were increased by Dox treatment in LN229/EGFRvIII cells (Supplementary Fig. 4d), suggesting that these phosphorylations of YTHDF2 are downstream of EGFR activation.

We have shown in Supplementary Fig. 4a that, the S39A/T381A double mutant completely disrupts ERK association, thus providing a direct evidence of the requirement of these two sites in ERK association. In the revision, we re-performed the same experiment with additional detection of pan-phosphothreonine and -phosphoserine of YTHDF2. It shows that YTHDF2 has ERK associated serine and threonine phosphorylation sites and S39 and T381 are the major ones in GSC11 cells (Supplementary Fig. 4a).

3. For the *in vivo* study of fig 2, Ki67 staining should be carried out on tumors to support differences observed in cell proliferation *in vitro*. Similarly, staining should be carried out for markers of cell death (Caspase3 or TUNEL). Staining for YTHDF2 could also be carried out to show loss of expression in tissue, as well as restored expression in the rescue cohort.

Response: Following the suggestions, we stained the brain sections from the *in vivo* study of Fig. 2 with Ki-67, cleaved Caspase-3 and YTHDF2 antibodies. The results from the experiments showed that YTHDF2 depletion decreased Ki-67 expression and increased cleaved Caspase-3 expression (Fig. 2e, f, Supplementary Fig. 2j). Also, the results from YTHDF2 IHC staining showed a loss of expression of YTHDF2 in shYTHDF2 tumors, but restored the expression in the YTHDF2 rescue tumors (Supplementary Fig. 2j).

4. Details should be provided concerning *in vitro* invasion assays.

Response: We added more details of *in vitro* invasion assays in the figure legend and Supplementary methods section in the revised manuscript.

5. Figure panels 2a, 2e, and 2f are more appropriate for supplemental materials.

Response: We have moved Figure panels 2a, 2e, and 2f to Supplementary Fig. 2a, 2d and 2h in the revised manuscript.

6. The impact of the S39A/T381A mutant in cell and tumor assays beyond *in vitro* invasiveness should be shown.

Response: We thank the reviewer for the important suggestion. We have investigated the impact of the S39A/T381A mutant in cell proliferation and tumor growth of YTHDF2-depleted GSC11 and GSC7-2 cells. We found that, as compared to wild-type YTHDF2, S39A/T381A mutant displayed minimal effects on the promotion of cell proliferation (Fig. 4d, e), and tumor growth (Fig. 4f, Supplementary Fig. 4c) of the YTHDF2-depleted GSC11 and GSC7-2 cells. These results suggest that S39/T381 sites are important to cell proliferation and tumor growth of GSC cells.

7. In fig 5 multiple distinct siRNAs should be employed in all studies for scientific rigor.

Response: We thank the reviewer for the suggestion. We have now employed two different siRNAs for each target gene to perform the new experiments for all studies in Fig. 6.

8. In fig 5 additional experiments should be carried out to demonstrate that the YTHDF2 m⁶A recognition mutant can rescue the EGF- or EGFRvIII-induced decrease in LXRA and HIVEP2 mRNA expression.

Response: Following the suggestion, we expressed the wild type and m⁶A recognition deficient YTHDF2 in YTHDF2 depleted GSC11 cells. We found that under the EGF treatment, expression of wild type YTHDF2 inhibited the mRNA expression of LXRA and HIVEP2 in the cells, but m⁶A recognition deficient YTHDF2 did not have the effect. We have added these results to Supplementary Fig. 6g.

9. Do the YTHDF2 phosphorylation site mutants affect LXRA and HIVEP2 transcript stability?

Response: In order to answer this question, we expressed wild type and phosphorylation sites mutated (S39A/T381A) YTHDF2 in YTHDF2 depleted GSC11 cells. We found that, as compared to wild type YTHDF2, phosphorylation sites mutated YTHDF2 significantly reduced the ability of increasing LXRA and HIVEP2 mRNA decay (Supplementary Fig. 6f). This is most likely due to that S39A/T381A mutant protein is unstable compared to the wild type YTHDF2 protein as demonstrated by the results in Fig. 4b.

10. In fig 6a, 6b, and 6c the authors should show the effect of the single knockdown alone, compared to the triple knockdown, as well as different permutations of double knockdowns to unravel individual contributions.

Response: Following the suggestion, we re-performed the proliferation and invasion assays in the figures with the single knockdown alone, and with comparison to the double knockdowns and the triple knockdowns. We found that in the presence of YTHDF2, a single knockdown of LXRA or HIVEP2 displayed no effect on cell invasion (Supplementary Fig. 7e), and HIVEP2 knockdown slightly increased cell growth (Supplementary Fig. 7c). However, in the absence of YTHDF2 (i.e. YTHDF2 depleted cells), a single knockdown of HIVEP2 significantly enhanced cell growth whereas LXRA knockdown enhanced both cell proliferation and invasion (Fig. 7b, c and Supplementary Fig. 7b). Furthermore, the double knockdown of LXRA or HIVEP2 fully rescue the growth and invasion defects caused by YTHDF2 depletion. These results suggest that these cells take advantage of the high YTHDF2 expression to maintain very low levels of LXRA or HIVEP2 for invasive growth. Also, because the cells with high YTHDF2 expression, like GSC11 cells, the level of LXRA is already very low, knockdown of LXRA would not cause any effect on cell growth and invasion. We have updated these results in Fig. 7a-c and Supplementary Fig. 7a-e).

11. Again in fig 6, scientific rigor dictates the use of multiple shRNA constructs for each target.

Response: Following the suggestion, we applied two shRNA constructs for LXRA and HIVEP2 in the experiments. The new results were included in Supplementary Fig. 7.

12. For the cholesterol experiments of fig 6, the effect on cellular cholesterol with knockdown of LXRA alone should be shown as a control.

Response: Following this suggestion, we re-performed the cholesterol experiments with knockdown of LXRA alone as control. We found that in the presence of YTHDF2, knockdown of LXRA displayed slightly increasing but not significant effect on cellular cholesterol (Fig. 7f), which might be due to the very low level of LXRA expressed in GSC11 cells. However, in the absence of YTHDF2 (by YTHDF2 knockdown), knockdown of LXRA rescued the decreasing of cellular cholesterol caused by YTHDF2 depletion (Fig. 7f). These results suggest that these cells take advantage of the high YTHDF2 expression to retain low level of LXRA for maintaining cellular cholesterol.

13. Figure 6k and l suggest that SSTR2 as a downstream target of HIVEP2 has antiproliferative effects that that YTHDF2 is able to regulate expression of SSTR2. The authors conclude that YTHDF2 regulates cell proliferation in this context, however, cell proliferation is never shown. These experiments need to be included.

Response: Following this suggestion, we investigated the role of SSTR2 in cell proliferation of YTHDF2-depleted GSC11 and GSC7-2 cells. We found that SSTR2 knockdown partially rescued the inhibition effect of YTHDF2 depletion on proliferation of the cells. The results have been updated in Supplementary Fig. 7m in the revised manuscript.

Responses to Reviewer #4

In this manuscript, Fang et. al. report an oncogenic role of m6A reader YTHDF2 in GBM for the first time. Together with their previous study on the function of m6A demethylase ALKBH5 in GBM (Zhang et. al., Cancer cell, 2017.), it reveals the aberrant of m6A modification and its regulatory machinery plays an important role in the tumorigenesis and development of GBM. The authors also showed that the EGFR/SRC/ERK signaling phosphorylates YTHDF2 and thus promotes YTHDF2 stability, which represents a novel mechanism regulating expression of YTHDF2 at the post-translational level. Overall, this is an interesting and timely study, and the paper is well written. Nevertheless, while the authors have provided extensive data to support the mechanism and function of YTHDF2 in GBM, there are some additional experiments and analyses that would further greatly strengthen the paper.

Response: We thank the reviewer for the positive comments.

1. The authors used S39A and T381A mutated YTHDF2 to demonstrate these two sites are important for ERK binding and YTHDF2 stability. However, the direct evidence of phosphorylation of S39 and T381 in GBM are missing. Besides, in order to further support the conclusion that ERK-mediated phosphorylation of YTHDF2 stabilizes YTHDF2, the authors could examine whether ERK inhibitor shorten YTHDF2 protein half-life in GBM cells.

Response: We thank reviewers for these valuable suggestions. Thus, to provide direct evidences of phosphorylation of S39 and T381 in GBM, we performed *in vitro* kinase experiments and *in vivo* experiments using pan-phosphothreonine or -phosphoserine antibody for detecting the phosphorylated YTHDF2.

In the *in vitro* kinase assay with recombinant active-ERK1, YTHDF2 wild-type and phosphor-mutants, we found that ERK1 phosphorylates YTHDF2-WT strongly, and S39A- and T381A-YTHDF2 to a lesser extent, but not S39A/T381A double mutant YTHDF2 (Fig. 4a). These results suggest that YTHDF2 can be phosphorylated by ERK directly and the S39 and T381 sites are the primary ERK phosphorylation sites.

Next, in the *in vivo* experiment using pan-phosphothreonine and -phosphoserine antibodies, we detected threonine and serine phosphorylation of YTHDF2 in GSC11 cells (Supplementary Fig. 4d). Moreover, the threonine and serine phosphorylations of YTHDF2 were suppressed by an EGFR inhibitor in the cells (Supplementary Fig. 4d). Also, these phosphorylations of YTHDF2 were increased by Dox treatment in LN229/EGFRvIII cells (Supplementary Fig. 4d), suggesting that these phosphorylations of YTHDF2 are downstream of EGFR activation.

Previously, we have shown in Fig. 4i that, the S39A/T381A double mutant completely disrupts ERK association, thus providing a direct evidence of the requirement of these two sites in ERK association. In the revision, we re-performed the same experiment with additional detection of pan-phosphothreonine and -phosphoserine of YTHDF2. It shows that YTHDF2 has ERK associated serine and threonine phosphorylation sites and S39 and T381 are the major ones in GSC11 cells (Supplementary Fig. 4a).

We have performed experiments to examine whether ERK inhibitor shorten YTHDF2 protein half-life in GBM cells, as suggested by the reviewer. The results from the experiments show that after inhibition of ERK using SCH772984, the half-life of YTHDF2 protein was shorten in LN229/EGFRvIII cells (Fig. 3g), supporting that ERK-mediated phosphorylation of YTHDF2 stabilizes YTHDF2.

2. The authors identified YTHDF2 targets by integrative analysis of RIP-seq and RNA-seq. Although 3,940 transcripts were associated with YTHDF2, only 19 genes among them were upregulated by YTHDF2 knockdown (KD). I think the number is relatively low as Wang et. al. reported that YTHDF2 had a significant and global impact on target RNA stability across the transcriptome (Wang et. al. Nature, 2014.). This is largely due to the small number (in total only 111) of differentially expressed transcripts identified by their RNA-seq data. It is better to redo RNA-seq with a higher reader coverage, and to ensure the KD efficiency of siRNAs (or it might be better to use shRNAs instead). Or, at least the authors can try combine RIP-seq and meRIP-seq first to identify the transcripts associated with YTHDF2 and also containing m6A sites as m6A-dependent target genes. Then analyze the expression changes of m6A-dependent target genes in global or in individual genes upon YTHDF2 KD.

Response: We thank the reviewer for the important suggestions. We have re-performed RNA-seq, and found that 1412 genes were differentially expressed and 730 genes were upregulated (1.5 folds) by YTHDF2 knockdown. Among these genes, both LXRA and HIVEP2 were upregulated. We also found that the downstream genes of LXRA were upregulated upon YTHDF2 knockdown (ABCA1 was upregulated by 1.50 folds and APOE was upregulated by 1.65 folds). Next, we followed the suggestion of the reviewer that “combine RIP-seq and meRIP-seq first to identify the transcripts associated with YTHDF2 and also containing m6A sites as m6A-dependent target genes” and identified 3518 transcripts as m6A-dependent target genes of YTHDF2. We have updated these results in Fig. 5b-d and Supplementary table 2, 3.

3. Besides m6A-recognition defective YTHDF2, the authors could also knockdown m6A methyltransferase METTL3 and/or METTL14 in GBM cells to demonstrate m6A-dependent regulation of LXRA and HIVEP2 by YTHDF2.

Response: Following the suggestion, we have knocked down m6A methyltransferase METTL14 in GSC11 cells as well as in YTHDF2-knockdown GSC11 cells, and then examined the levels of LXRA and HIVEP2 in the cells. We found that upon METTL14 knockdown, YTHDF2 knockdown did not affect the mRNA expression of LXRA and HIVEP2 (Supplementary Fig. 6e), suggesting that YTHDF2 regulates the mRNAs of LXRA and HIVEP2 in an m⁶A dependent manner.

4. In addition to the activation by EGFR/SRC/ERK, mutations in YTHDF2 amino acid sequence might also affect the stability of YTHDF2. For example, the mutation of lysine might prevent ubiquitination and degradation of YTHDF2 protein. Please check the mutation data in GBM and discuss.

Response: We thank the reviewer for the suggestion. We have queried TCGA data for mutations in YTHDF2 through cBioPortal FOR CANCER GENOMICS. We found that among 507 lower grade glioma and 378 GBM patients, only two patients were identified by YTHDF2 mutation (one patient with V505I mutation and one with R527W mutation). Furthermore, we also queried other datasets, and found that the amino acids of YTHDF2 are rarely mutated in GBM. We added the information to the section of Discussion in the revised manuscript.

REVIEWERS' COMMENTS

Reviewer #1 (Remarks to the Author):

The authors have answered all my questions. I recommend publication.

Reviewer #2 (Remarks to the Author):

The manuscript has been significantly improved. The work is interesting and worthy to be published in Nature Communications.

Reviewer #4 (Remarks to the Author):

In this revised manuscript, the authors have conducted a series of new experiments and added additional data, and have carefully and thoroughly addressed all the critiques from the reviewers. The manuscript has been substantially revised. As a result, the quality of this paper has been further improved. Overall, this is an interesting and timely study. Therefore, I fully support the publication of this paper in Nature Communications.

Responses to Reviewers' Comments

We thank the reviewers for thorough evaluation of our original manuscript. The comments and suggestions are highly constructive and genuinely helpful. We thank the reviewer for acknowledging the importance and strength of our work.

Reviewer #1 (Remarks to the Author):

The authors have answered all my questions. I recommend publication.

Responses: No further critiques.

Reviewer #2 (Remarks to the Author):

The manuscript has been significantly improved. The work is interesting and worthy to be published in Nature Communications.

Responses: No further critiques.

Reviewer #4 (Remarks to the Author):

In this revised manuscript, the authors have conducted a series of new experiments and added additional data, and have carefully and thoroughly addressed all the critiques from the reviewers. The manuscript has been substantially revised. As a result, the quality of this paper has been further improved. Overall, this is an interesting and timely study. Therefore, I fully support the publication of this paper in Nature Communications.

Responses: No further critiques.